# Direct visualization of stacking-selective self-intercalation in epitaxial Nb$_{1+x}$Se$_2$ films

Hongguang Wang [1] ✉, Jiawei Zhang [1], Chen Shen [2] ✉, Chao Yang [1], Kathrin Küster[1], Julia Deuschle[1], Ulrich Starke [1], Hongbin Zhang[2], Masahiko Isobe [1], Dennis Huang [1] ✉, Peter A. van Aken [1] & Hidenori Takagi [1,3,4]

Two-dimensional (2D) van der Waals (vdW) materials offer rich tuning opportunities generated by different stacking configurations or by introducing intercalants into the vdW gaps. Current knowledge of the interplay between stacking polytypes and intercalation often relies on macroscopically averaged probes, which fail to pinpoint the exact atomic position and chemical state of the intercalants in real space. Here, by using atomic-resolution electron energy-loss spectroscopy in a scanning transmission electron microscope, we visualize a stacking-selective self-intercalation phenomenon in thin films of the transition-metal dichalcogenide (TMDC) Nb$_{1+x}$Se$_2$. We observe robust contrasts between 180°-stacked layers with large amounts of Nb intercalants inside their vdW gaps and 0°-stacked layers with little detectable intercalants inside their vdW gaps, coexisting on the atomic scale. First-principles calculations suggest that the films lie at the boundary of a phase transition from 0° to 180° stacking when the intercalant concentration $x$ exceeds ~0.25, which we could attain in our films due to specific kinetic pathways. Our results offer not only renewed mechanistic insights into stacking and intercalation, but also open up prospects for engineering the functionality of TMDCs via stacking-selective self-intercalation.

Stacking and intercalation represent two of the most versatile tuning knobs for modifying and engineering the properties of two-dimensional (2D) van der Waals (vdW) materials[1–3]. Stacking as a control parameter includes varying the lateral registry or twist angle between adjacent layers, to realize, for example, exotic electronic phases on a moiré superlattice[4,5]. Layers from different compounds can also be stacked to fabricate heterostructures to achieve functionalities such as atomically sharp *p-n* junctions and interlayer excitons[6,7]. Intercalation involves inserting atoms or molecules into the vdW gap between two layers, either by synthetic routes or by electrochemistry[8–11]. The intercalants serve a manifold role, either as

stored ions for battery applications[10], or as a means to transfer additional charge carriers or tune interlayer coupling, possibly resulting in new superconducting[12,13] or magnetic states[14,15]. While the individual potentials of stacking and intercalation have hardly been exhausted, a natural extension is to explore synergistic combinations of these two tuning parameters. One possibility is to utilize intercalation to stabilize alternative stacking arrangements. For example, Li intercalation of MoS$_2$ changes the structure from a 2$H_c$ polytype with trigonal prismatic coordination of the Mo atoms and 180° rotation between successive layers to a 1$T$ polytype with octahedral coordination of the Mo atoms and 0° rotation between successive layers[16]. Another possibility

[1]Max Planck Institute for Solid State Research, Heisenbergstr. 1, 70569 Stuttgart, Germany. [2]Department of Materials and Earth Sciences, Technical University of Darmstadt, Darmstadt, Germany. [3]Institute for Functional Matter and Quantum Technologies, University of Stuttgart, 70569 Stuttgart, Germany. [4]Department of Physics, University of Tokyo, 113-0033 Tokyo, Japan. ✉e-mail: hgwang@fkf.mpg.de; chenshen@tmm.tu-darmstadt.de; D.Huang@fkf.mpg.de

is to utilize stacking sequences to selectively intercalate between layers or regions of interest. For example, in a heterostructure composed of graphene and hexagonal boron nitride (hBN) layers, Li atoms intercalate the graphene-graphene and graphene-hBN interfaces, while avoiding the hBN-hBN interfaces, resulting in a layer-selective intercalation[17,18].

Transition-metal dichalcogenides (TMDCs) are particular among the 2D vdW materials in that they exhibit a self-intercalation phenomenon. In the synthesis of $MX_2$, where $M$ is a transition metal atom and $X =$ S, Se, or Te, an excess starting amount of $M$ may either result in $X$ vacancies, or $M$ atoms incorporated into the vdW gap[19–22]. Early studies showed that TMDCs, especially with group-5 metal atoms, i.e., $M =$ V, Nb, or Ta, undergo complex transformations of stacking polytypes upon excess $M$ stoichiometry[23,24]. For example, bulk samples of $Nb_{1+x}S_2$ and $Nb_{1+x}Se_2$ undergo a transition from a $2H_a$ polytype with 180° stacking of layers to a $3R$ polytype with 0° stacking when $x$ crosses a threshold around 0.03–0.07, presumably because the $3R$ polytype better accommodates the increasing number of Nb intercalants in its interstitial voids[24–27]. With larger values of $x$, polycrystalline samples with $2H_a$, $2H_b$, and $3R$ polytypes and Nb intercalants have been stabilized[24], but less is known about this regime. In essence, the cooperative behavior of stacking and intercalation is already realized "naturally" in the rich chemistry and polytypism of $M_{1+x}X_2$, but the mechanistic details remain hidden, and the behavior remains to be exploited for applications. To determine the precise atomic position and chemical state of the intercalants in a mixed-phase sample with various stacking configurations, cross-sectional scanning transmission electron microscopy (STEM) with sub-angstrom resolution is ideal[28–36] and avoids the problem of interlayer averaging present in other techniques, such as x-ray diffraction[24–27] or Raman spectroscopy[37,38]. To exploit the cooperative behavior of stacking and intercalation for technological purposes, it is useful to reproduce the previously synthesized $M_{1+x}X_2$ compounds as thin films, which offer large surface areas, precise thickness control, amenability to lithography and plentiful pathways for properties engineering[32,39–42]. A comparative study using thin films may also elucidate the role of growth kinetics in tuning stacking and intercalation.

We employ a thin-film approach, together with STEM and first-principles calculations, to offer a fresh perspective on the $Nb_{1+x}Se_2$ system. With the high spatial resolution afforded by STEM, we observe that thin films with an average $x \sim 0.29$ comprise a nanoscale phase mixture of $NbSe_2$ layers stacked with both 180° and 0° in-plane rotations. The 180°-stacked layers are highly self-intercalated with Nb at the octahedral interstitial sites, with likely several tens of percentage occupancy, whereas the 0°-stacked layers have few resolvable intercalants at their octahedral interstitial sites. Our results go beyond simply imaging intercalants or different stacking structures to establishing a correlative relationship between the two. We further evaluate the roles of thermodynamics and kinetics in these observations. Density functional theory (DFT) confirms that the energetically favorable stacking orientation shifts from 0° back to 180° when the self-intercalation exceeds a threshold $x \sim 0.25$. To reach this threshold, however, kinetic pathways that are distinct from the thin films appear to be crucial. Attempts to replicate these observations with $Nb_{1+x}Se_2$ crystals grown via chemical vapor transport (CVT) result in a homogeneous phase of intercalated, 0°-stacked layers with an average $x$ of at most $\sim 0.20$. The Nb intercalants reduce the size of the Fermi hole pockets and suppress superconductivity in $NbSe_2$, and these modified properties may form the basis for fabricating junctions and nanostructures upon achieving more precise control of stacking and self-intercalation.

## Results

### Basic characterization

Thin films of $Nb_{1+x}Se_2$ were deposited on $c$-cut sapphire substrates via a "hybrid" pulsed laser deposition technique (hPLD)[43,44]. Figure 1a

illustrates the working principle of hPLD, which combines the laser ablation of refractory Nb metal with thermal sublimation of low-vapor-pressure Se to achieve adsorption-controlled growth, similar to that in molecular beam epitaxy (additional details are given in the Methods section). The growth process was monitored in situ by reflection high-energy electron diffraction (RHEED). During film deposition, the substrate spots disappear as new streaks emerge, indicating the formation of 2D, crystalline $Nb_{1+x}Se_2$ (Fig. 1b, c) (see Supplementary Fig. 1 for additional RHEED patterns and Supplementary Table 1 for growth parameters). Scanning tunneling microscopy (STM) (Fig. 1d) reveals a hexagonal lattice of bright spots corresponding to the topmost Se atoms on the surface $NbSe_2$ layer (Fig. 1e). The topography shows clear background inhomogeneity, an effect which we later attribute to an inhomogeneous distribution of subsurface Nb intercalants[45] (see Supplementary Fig. 2 for additional STM images). High-angle annular dark-field (HAADF) STEM images (Fig. 1f, g) show the cross-section of an 8–9 layer thick film, sandwiched between the sapphire substrate and a protective Se capping layer (see Supplementary Fig. 3 for additional STEM images in different magnifications). Each $NbSe_2$ layer, arranged as a Se-Nb-Se triple layer, can be fully visualized with atomic resolution. When viewing along the [1–100] direction of sapphire, we mainly observed two kinds of cross sections of the triple-layer structure: (1) Two Se atoms located directly above and below each other, but shifted laterally with respect to the Nb atom (Fig. 1f). (2) Two Se atoms located directly above and below each other, and also vertically aligned with the Nb atom (Fig. 1g). The first cross-section corresponds to the [11–20] projection of $NbSe_2$ and confirms that the individual $NbSe_2$ layers adopt the metallic $1H$ phase with trigonal prismatic coordination of the Nb atoms, whereas the second cross-section corresponds to the [1–100] projection of $NbSe_2$ (see Fig. 1e). These two distinct cross sections imply that the $Nb_{1+x}Se_2$ film is composed of domains rotated primarily by 30° with respect to each other, which is typical of our samples (see Supplementary Fig. 4 for low-energy electron diffraction (LEED) patterns of additional films). When we cut along the [11–20] direction of sapphire, we again observe these two kinds of $NbSe_2$ cross sections, which confirms that they correspond to 30°-rotated domains (see Supplementary Fig. 5).

### Stacking-selective self-intercalation

A close-up HAADF-STEM image of a film region (Fig. 1h) not only reveals each Nb and Se atom column with exquisite resolution but also manifests an additional row of atoms intercalated between the vdW gap of two $NbSe_2$ triple layers. The intercalant atoms are vertically aligned with the Nb atoms above and below and appear at every site, although with different intensities. The varying brightness at each intercalation site indicates varying concentrations of intercalants because the HAADF-STEM image is a cross-sectional average and its signal intensity is proportional to the mass of materials. Both center-of-mass and 2D Gaussian fitting methods are applied to precisely determine the positions of each atomic column in STEM images (see Supplementary Fig. 6 on the quantification of simulated STEM image of $2H_a$-$NbSe_2$ as a reference). We observe clear, local structural modifications induced by the intercalant row, including an expansion of the out-of-plane spacing between adjacent $NbSe_2$ layers (Fig. 1i). (XRD measurements over a macroscopic area consisting of many intercalated regions confirm an overall increase in the average interlayer spacing, consistent with previous polycrystal studies of $Nb_{1+x}Se_2$[24]; see Supplementary Fig. 7.) There is also an increase in the Se-Nb-Se bond angle for the two $NbSe_2$ layers surrounding the row of intercalants and a third $NbSe_2$ layer above (Fig. 1j).

To determine the chemical identity of the intercalants, we performed atomic-resolution electron energy-loss spectroscopy (EELS) for the cross-section shown in Fig. 2. The EELS spectra (Fig. 2a, b) reveal the Se-$L_{2,3}$ edge at the Se capping layer (1) and the Nb-$L_{2,3}$ edge at the intralayer Nb site (4) and intercalation site (3). A weak Se-$L_{2,3}$

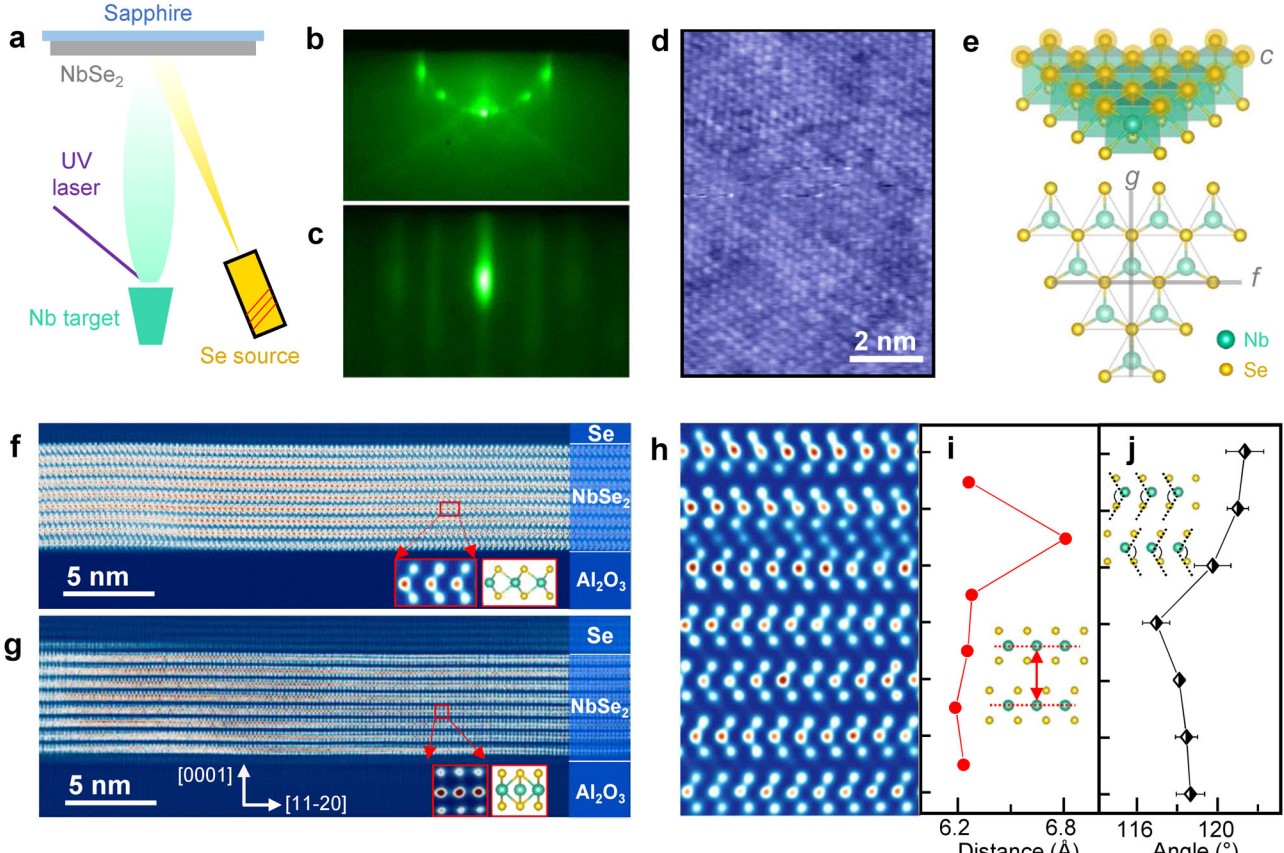

**Fig. 1 | Basic atomic-scale characterization. a** Schematic of the hybrid pulsed laser deposition (hPLD) growth of $Nb_{1+x}Se_2$ thin films. **b, c** Reflection high-energy electron diffraction patterns at different stages of the growth: (**b**) sapphire substrate before deposition and (**c**) $Nb_{1+x}Se_2$ film after 17,700 laser pulses; direction [1–100]; electron energy 10 keV. **d** Atomically resolved scanning tunneling microscopy topographic image; setpoint −125 mV, 50 pA. The bright spots correspond to surface Se atoms. **e** Crystal structure of a $NbSe_2$ layer shown in two perspectives. The letter "c" labels the top-layer Se atoms (shaded in orange) seen in (**c**). The lines marked by the letters "f" and "g" depict the view along which the cross-sections in

(**f**) and (**g**) are seen. **f, g** High-angle annular dark-field scanning transmission electron microscopy (HAADF-STEM) images of the cross-section of a $Nb_{1+x}Se_2$ ($x ~ 0.29$) thin film viewed along the [1–100] direction of the sapphire substrate, corresponding to two distinct film domains with 30° rotation (cuts shown in **e**). The insets show intralayer lattice structure and their structural models. **h–j** A close-up HAADF-STEM image revealing the $Nb_{1+x}Se_2$ lattice structure (**h**) and the calculated interlayer spacing (**i**) and Se-Nb-Se angle (**j**) averaged over each $NbSe_2$ layer, as illustrated by the insets. The error bars correspond to the standard deviations of the Se-Nb-Se angles in each $NbSe_2$ layer.

signal is present in spectra (2), (3), and (4), presumably due to de-channeling and the ensuing beam broadening of neighboring Se atomic planes[46,47]. EELS spectrum images of the Al-K, Se-$L_{2,3}$, and Nb-$L_{2,3}$ edges (Fig. 2c–e) and their composite map (Fig. 2f) show the distribution of constituent elements at the atomic scale (see Supplementary Fig. 8 for extracted EELS spectra). The row-averaged signal profiles (Fig. 2g) show maxima in the Nb signal corresponding to the intralayer Nb planes and intercalated layers. The intercalants are unambiguously established to be Nb atoms, thus pointing to a self-intercalation phenomenon in our $Nb_{1+x}Se_2$ films. We estimate based on x-ray photoelectron spectroscopy (XPS) that on average, the global $x$ value of our thin films is roughly 0.29 (see Supplementary Figs. 9, 10, and Supplementary Table 2). Further information can be derived by spatial mapping at the Nb-$N_{2,3}$ edge at even lower energies (Fig. 2h–j). The EELS spectra (Fig. 2i) reveal small but systematic differences between the intralayer Nb and the Nb intercalants: For the intercalants, the onset, volume, and peak of the Nb-$N_{2,3}$ edge are shifted to lower energies (red, green, and black arrows in Fig. 2i, which can also be visualized in the first derivative of the line scan maps (Fig. 2j). Both the intralayer Nb and the Nb intercalants are cationic[48], but the shift to lower energies for the Nb intercalants indicates that they have a slightly reduced positive valence state and are more weakly bonded to the neighboring Se

atoms. This observation highlights the extreme sensitivity of STEM-EELS in resolving minute chemical differences at the atomic scale.

Figure 3 encapsulates the main finding of this paper, which is a stacking-selective self-intercalation in our $Nb_{1+x}Se_2$ films with an average $x ~ 0.29$. Four HAADF-STEM images are presented (Fig. 3a–d). The films show layers that are stacked with either 180° and 0° in-plane rotation, as indicated by the red and blue arrows. We find that different lateral regions of the film show predominant 180° (Fig. 3a, b) or 0° stacking (Fig. 3c, d), but within each such region, there are also changes in stacking rotation from layer to layer. The 180°-stacked layers have the same lateral registry as the $2H_a$ polytype of bulk, pristine $NbSe_2$ (Fig. 3e). The Nb atoms are all vertically aligned, but since each layer rotates by 180°, a unit cell consists of two $NbSe_2$ layers. The stacking motif of Nb atoms is AA′, where the two A's indicate the identical lateral position of Nb atoms, and the prime indicates a 180° rotation. For the 0°-stacked layers, we find that the Nb atoms in adjacent layers undergo a lateral shift. In the top four layers of Fig. 3c, the stacking motif is AB, meaning that the Nb atoms of every second $NbSe_2$ layer are realigned vertically (Fig. 3f). This corresponds to a known but rare $2H_b$ polytype of $Nb_{1+x}Se_2$[24,49]. In Fig. 3d and the bottom four layers of Fig. 3c, we observe that the periodicity is four $NbSe_2$ layers, with stacking motif ABAC (Fig. 3g). According to the Ramsdell notation, the corresponding polytype is $4H$ (4 layers per unit cell, hexagonal

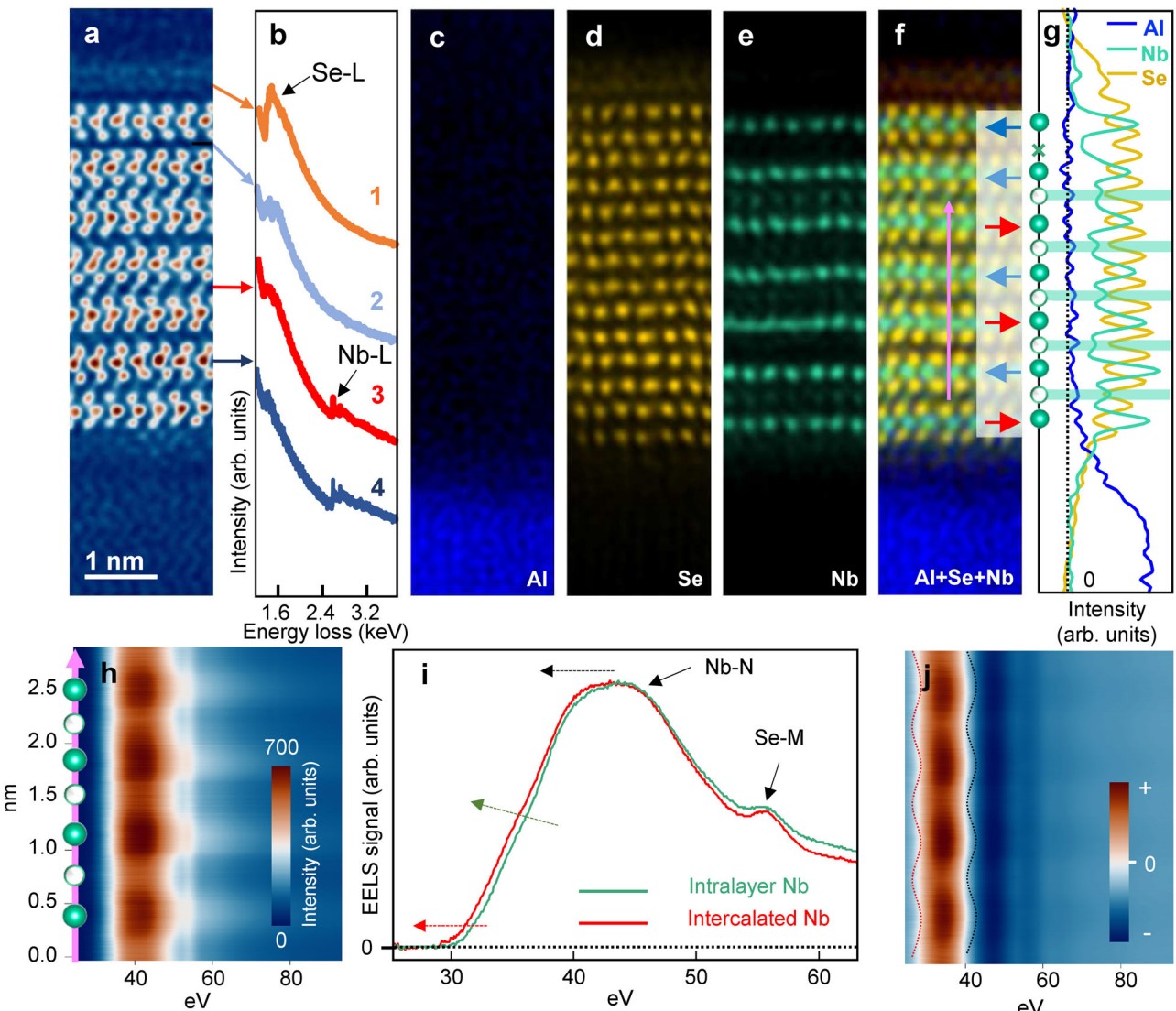

**Fig. 2 | Electron energy-loss spectroscopy (EELS) measurement of the Nb$_{1+x}$Se$_2$ ($x \sim 0.29$) thin film. a** HAADF-STEM image and (**b**) EELS spectra at four regions: (1) Se capping layer, (2) a vdW gap without intercalation, (3) a vdW gap with intercalants, (4) the Nb plane. **c, d, e** correspond to the spectrum images of the Al-K, Se-L$_{2,3}$, and Nb-L$_{2,3}$ edges. **f** and **g** are the composite map of Al, Se, Nb and corresponding signal intensity profiles. The blue and red arrows in (**f**) mark the different in-plane orientations of NbSe$_2$ layers. Shaded areas in (**g**) mark the vdW gaps with intercalants. The vertical dashed line denotes the zero signal intensity. **h** is the background-subtracted EELS line scan at the Nb-N$_{2,3}$ edge taken along the vertical pink arrow in (**f**), perpendicular to the NbSe$_2$ layers. **i** EELS spectra at an intralayer Nb site and a Nb intercalant site. The red, green, and black arrows represent the red-shift of the onset, volume, and peak of the Nb-N$_{2,3}$ edge for the intercalated Nb. **j** is the first derivative of (**h**). The changes of the onset and peak positions of the Nb-N$_{2,3}$ edge are highlighted by the dotted red and black curves, respectively.

symmetry), but what we observe does not correspond to any of the known 4$H$ polytypes (4$H_a$, 4$H_b$, 4$H_c$, 4$H_{dI}$, and 4$H_{dII}$[49]). We tentatively label the polytype as 4$H_x$, and highlight the utility of STEM in resolving such fine differences between 2$H_b$ and 4$H_x$ polytypes with 0° stacking, which would be otherwise difficult to distinguish with XRD. We note that in Fig. 1h, the 0°-stacked layers show a stacking motif of ABC, which corresponds to the 3$R$ polytype of NbSe$_2$. To summarize, we observed a total of four polytypes: 2$H_a$-AA', which we refer to as a 180°-stacked polytype, and 2$H_b$-AB, 3$R$-ABC, and 4$H_x$-ABAC, which we collectively refer to as 0°-stacked polytypes.

The striking point in Fig. 3 is not only the stacking variation in our Nb$_{1+x}$Se$_2$ films, but the selective preference of Nb intercalants for the 180°-stacked regions. Here, the Nb intercalants are always vertically aligned with the Nb atoms in adjacent layers, meaning that they occupy the octahedral site in the vdW gap (Fig. 3e). Meanwhile, there is little detectable concentration of intercalants in the 0°-stacked regions. The contrast is stark when comparing Fig. 3a and Fig. 3b with

predominant 180° stacking and many intercalants to Fig. 3c, d with predominant 0° stacking and very few intercalants. The difference is also evident in the interlayer spacing, which expands with the presence of Nb intercalation and increases for greater concentration of intercalants (Fig. 3a–d). Sharp contrasts in intercalant concentration are also evident on a layer-by-layer basis. In Fig. 3b, all the pairs of layers except one have 180° stacking and many intercalants, and the only pair with 0° stacking has a barely intercalated vdW gap with reduced interlayer spacing. In Fig. 3c, all the pairs of layers except one have 0° stacking and little intercalants, and the only pair with 180° stacking shows a clear row of intercalants with enhanced interlayer spacing. Again, in Fig. 1h, the vdW gaps between 0°-stacked layers are barely intercalated, whereas the single vdW gap with a significant presence of intercalants lies between two 180°-stacked layers. (See Supplementary Fig. 11 for an example of a lateral transition from 180° stacking with many intercalants to 0° stacking with little intercalants.)

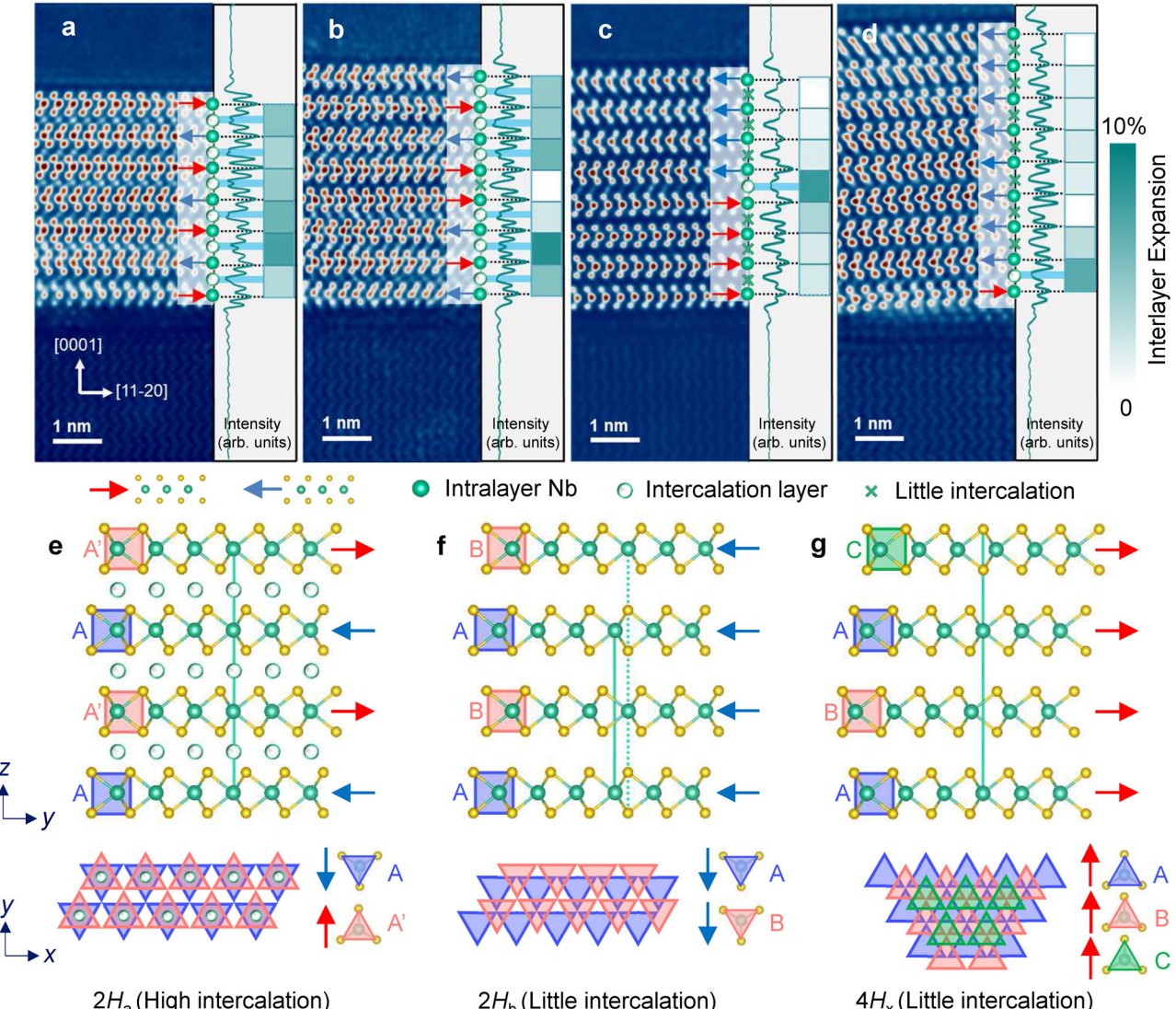

**Fig. 3 | Stacking-selective self-intercalation in Nb$_{1+x}$Se$_2$ ($x \sim 0.29$). a–d** Atomically resolved HAADF-STEM images of the epitaxial Nb$_{1+x}$Se$_2$ thin film, showing different stacking sequences between NbSe$_2$ layers. The arrows mark the in-plane orientation of each NbSe$_2$ layer. The curves in the right panel are laterally integrated signal profiles along the [0001] growth direction. The horizontal dashed lines and light blue shaded areas mark the intralayer Nb planes and the vdW gaps with intercalants. The intensities of the blue squares indicate the amount of expansion of the interlayer spacing with reference to the value of a NbSe$_2$ crystal. **e, f, g** correspond to observed structures in the Nb$_{1+x}$Se$_2$ thin film: (**e**) 2$H_a$ phase with a high degree of intercalation; (**f**) 2$H_b$ phase with little intercalation, (**g**) 4$H_x$ phase with little intercalation. The vertical solid and dashed green lines illustrate the interlayer stacking sequences. Each structure is shown in two perspectives, and in the bottom structures, the NbSe$_6$ trigonal prisms are represented by triangles.

Within regions with many successive rows of intercalants between 180°-stacked NbSe$_2$, such as Figs. 2a and 3a with five and six successive rows of intercalants, respectively, we observed signs of staggering. In Fig. 2a, the first, third, and fifth vdW gaps away from the sapphire substrate have a larger concentration of intercalants, whereas the second and fourth vdW gaps have a lower concentration of intercalants. In Fig. 3a, the second, fourth, and sixth vdW gaps away from the sapphire substrate have a larger concentration of intercalants, whereas the first, third, and fifth vdW gaps have a lower concentration of intercalants. This alternation in intercalant concentration can also be seen in the row-averaged ADF signal profiles of Figs. 2a and 3a, where the local maxima corresponding to the Nb intercalants alternate in peak height.

## Electronic structure and properties

With a view toward possible control of electronic structure by intercalants, we investigated how the Nb intercalants modify the well-known electronic structure and properties of NbSe$_2$. Figure 4a, b present angle-resolved photoemission spectroscopy (ARPES) intensity cuts of a Nb$_{1+x}$Se$_2$ film along Γ-$M$ and Γ-$K$-$M$. The positions of the bands closest to the Fermi energy were extracted from energy and momentum distribution curves and are marked as circles. Even when probing a small area of the sample with a diameter of 10–30 μm, we unavoidably averaged over 180°- and 0°-stacked layers with high and low amounts of Nb intercalants, as well as some rotated domains. To model the inhomogeneous film with DFT, we utilized a Nb$_5$Se$_8$ slab with four NbSe$_2$ layers and a vacuum region (Fig. 4c). The inner two layers have 180° stacking with 100% Nb intercalation, whereas the outer two layers have 0° stacking and no Nb intercalants, which is a simplification of the actual situation. The overall structure retains 1 × 1 in-plane periodicity, so no unfolding of the bands is required, and the overall concentration of Nb intercalants, $x = 0.25$, is relatively close to the experimental estimate of ~0.29. Figure 4d, e shows the calculated bands projected onto the pristine (blue) and 100%-intercalated (red) layers of the slab,

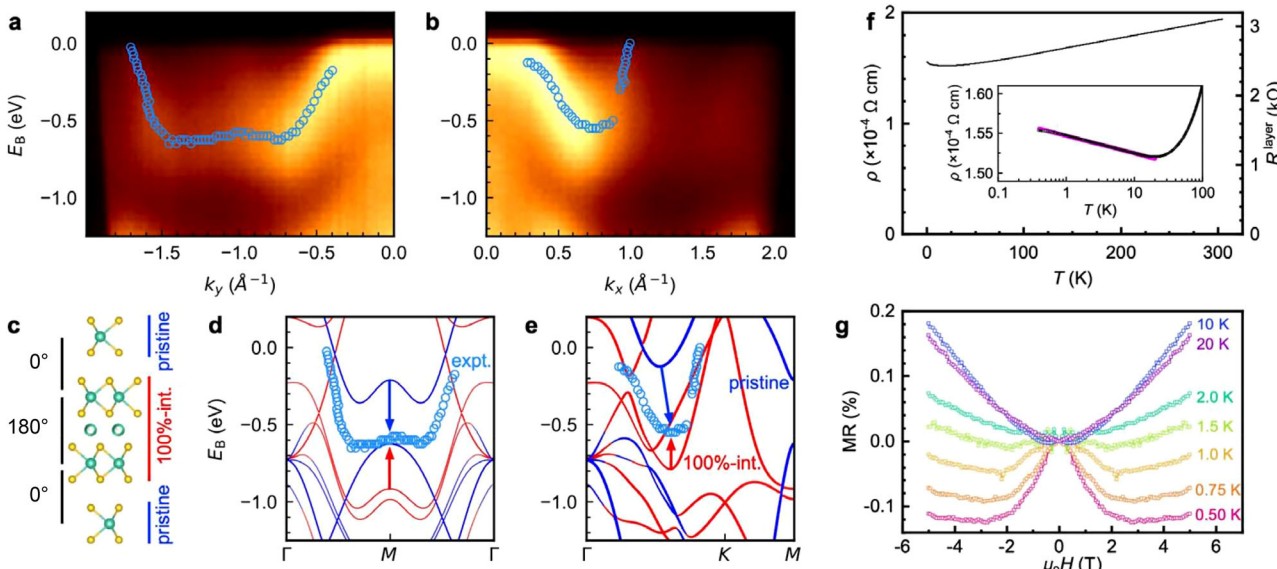

**Fig. 4 | Electronic structure and properties of Nb$_{1+x}$Se$_2$ ($x \sim 0.29$). a, b** Angle-resolved photoemission spectroscopy (ARPES) intensity cuts along momenta $k_y$ and $k_x$, which correspond to the $\Gamma-M-\Gamma$ and $\Gamma-K-M$ directions of the Brillouin zone, respectively. Due to the rotational disorder of the film, the intensity cuts show some admixture of bands along other directions. The extracted band maxima are overlaid as open circles. **c** Four-layer Nb$_5$Se$_8$ slab used to model the ARPES data, consisting of 100%-intercalated and pristine layers. **d, e** The band structure computed by density functional theory (DFT) is shown along the momentum directions corresponding to (**a**) and (**b**). The bands are projected onto the pristine (blue) and 100%-intercalated (red) layers. The arrows show the relative shift of the actual experimental bands compared to the calculations. **f** Resistivity ($\rho$) vs. temperature ($T$). Right axis: Sheet resistance per NbSe$_2$ layer. Inset: log($T$)-like upturn at low temperatures. The magenta straight line is a guide to the eye. **g** Magnetoresistance (MR) below 20 K. The magnetic field ($\mu_0 H$) is applied perpendicular to the plane of the film.

as well as the ARPES-derived bands near the Fermi energy. The calculated blue bands for the pristine NbSe$_2$ layers form Fermi-hole pockets at $\Gamma$ and $K$. The corresponding red bands for the intercalated NbSe$_2$ layers exhibit a downward shift as large as 0.8 eV, implying effective electron doping. The ARPES bands lie between these two limits of zero and 100% intercalation. The donation of electrons from Nb intercalants is consistent with their cationic valence state seen by EELS (Fig. 2i).

Electrical transport measurements of a Nb$_{1+x}$Se$_2$ film reveal metallic behavior with no superconductivity down to temperatures as low as 0.4 K and current densities as low as 3.8 A/mm$^2$, which is in contrast to the superconductivity observed in pristine NbSe$_2$ crystals and films (Fig. 4f). Below 20 K, the resistivity shows an upturn with logarithmic temperature dependence (inset of Fig. 4f). The upturn likely corresponds to weak localization, as a concomitant low-field negative magnetoresistance emerges below 20 K (Fig. 4g) (see Supplementary Fig. 12 for conductivity analysis). The sheet resistance per layer of the intercalated film lies between 2.4 and 3.1 kΩ over the measured temperature range. These values fall below but are close to the quantum resistance $h/(2e)^2 \sim 6.5$ kΩ[50], which is often the critical value above which a superconductor-to-insulator transition is triggered in a 2D system due to Cooper pair localization. The suppression of superconductivity could be in part related to the disorder induced by Nb intercalants. Furthermore, electron transfer from Nb intercalants reduces the size of hole pockets and lowers the density of states at the Fermi energy, which in turn decreases any superconducting transition temperature. The Nb intercalants could possibly also be magnetic and act as pair-breaking scatterers[32].

## Discussion

Our STEM images disclose that epitaxial films of Nb$_{1+x}$Se$_2$ near a global $x \sim 0.29$ exhibit a nanoscale separation into 180°-stacked layers with a large number of Nb intercalants, perhaps several tens of percentage occupancy (considering the average $x$ value), and 0°-stacked layers whose number of Nb intercalants falls close to the detection limit. Bulk studies have shown that the polytype of pristine NbSe$_2$ is 2$H_a$ with 180° stacking, and that a few percent of excess Nb should stabilize the 3$R$ polytype with 0° stacking[24–27]. Here, the situation seems to be reversed, with the 180°-stacked layers hosting a much greater number of Nb intercalants than the 0°-stacked layers. We seek to address (1) whether there could be a second polytype transition near $x \sim 0.29$ that favors 180° stacking again over 0° stacking, and (2) what role kinetics and growth conditions might play in stabilizing our thin film structures.

We performed a comparative study of bulk single crystals of Nb$_{1+x}$Se$_2$ grown by CVT. Figure 5 presents a series of HAADF-STEM images for four Nb$_{1+x}$Se$_2$ crystals with increasing average values of $x$. At $x = 0$, NbSe$_2$ crystallizes in the 2$H_a$ polytype with 180° stacking, and the vdW gaps are clear of intercalants (Fig. 5a). At $x \sim 0.11$, the NbSe$_2$ layers uniformly adopt the 3$R$ configuration with 0° stacking (Fig. 5b). There are faint signatures of some intercalants inside the octahedral voids of the vdW gaps, perhaps to a degree that is qualitatively similar to what we observe between the 0°-stacked layers of our thin films.

As we continued to synthesize crystals with increasing $x$ via CVT, the degree of self-intercalation began to saturate. An initial Nb:Se stoichiometry of 1.25:2 resulted in Nb$_{1+x}$Se$_2$ crystals with average $x \sim 0.17$, showing again a homogeneous 0°-stacked phase, but with more intercalants inside their octahedral voids (Fig. 5c). A greater starting Nb:Se stoichiometry of 1.35:2 resulted in Nb$_{1+x}$Se$_2$ crystals with an average $x \sim 0.20$. These crystals contained domains of additional Nb-Se phases (see Supplementary Fig. 13); nevertheless, in the regions with 1$H$ layers of NbSe$_2$, we still observed homogeneous 0° stacking (Fig. 5d). The degree of intercalation between the 0°-stacked layers here seems to be much greater than the 0°-stacked layers of our thin films (e.g., Figs. 1h or 3c), but still less than the 180°-stacked layers of our thin films (e.g., Fig. 3a). We did not observe any highly intercalated 180°-stacked layers in the bulk samples, even locally.

Along with the comparison of Nb$_{1+x}$Se$_2$ prepared via alternative growth techniques, we also performed DFT calculations, which can predict the equilibrium structures with the lowest ground-state

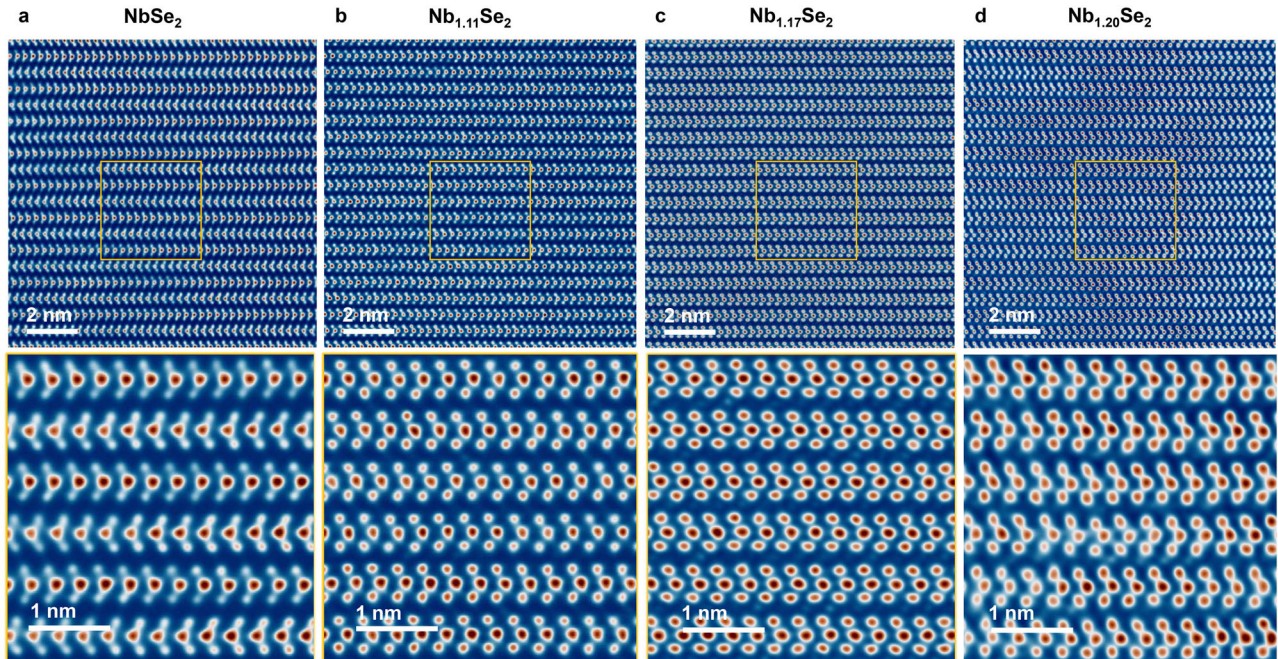

**Fig. 5 | Self-intercalation in single-crystal Nb$_{1+x}$Se$_2$ grown via chemical vapor transport. a–d** HAADF-STEM images of the cross-section of bulk Nb$_{1+x}$Se$_2$ with average $x = 0$, 0.11, 0.17, and 0.20, respectively. A close-up of the region enclosed by the yellow box is shown for each panel.

energy. We constructed supercells of bilayer Nb$_{1+x}$Se$_2$ with 180° and 0° stackings with Nb intercalants inside the octahedral voids. The supercells had chemical formulae Nb$_2$Se$_4$, Nb$_{19}$Se$_{36}$, Nb$_9$Se$_{16}$, Nb$_{10}$Se$_{16}$, and Nb$_3$Se$_4$, which correspond to an occupancy of the interstitial octahedral sites of 0%, 11%, 25%, 50%, and 100%. (For bilayer Nb$_{1+x}$Se$_2$ surrounded by vacuum, there is only one octahedral void per unit cell, so the occupancy of octahedral voids is given by $2x$, in contrast to bulk Nb$_{1+x}$Se$_2$, where the occupancy is given by $x$.) At 0% intercalation, the total energy after structural relaxation of 180°-stacked bilayer NbSe$_2$ is only 6 meV per formula unit smaller than that of 0°-stacked bilayer NbSe$_2$ (Fig. 6a). The near degeneracy of these two stackings reflects the weak interlayer coupling of vdW nature in NbSe$_2$, but the prediction of a slightly more energetically favorable 180° stacking is in line with the observed 2$H_a$ polytype in bulk, pristine NbSe$_2$. At 11% and 25% intercalations, the 0°-stacked bilayer is actually lower in energy than the 180°-stacked bilayer. The situation is then reversed at 50% and 100% intercalations, where the 180°-stacked bilayer is again more stable. DFT indeed predicts two polytype transformations with increasing $x$.

We also computed the intercalation energies in Fig. 6b, defined as $\Delta E_{int} = [E(Nb_{1+x}Se_2) - E(NbSe_2) - xE(Nb)]/x$, where $E(Nb_{1+x}Se_2)$ and $E(NbSe_2)$ are the DFT total energies after structural relaxation of the intercalated and pristine structures and $E(Nb)$ is the DFT-computed energy required to take one Nb atom from bulk Nb. In all but one case, $\Delta E_{int}$ is negative, meaning that self-intercalation actually releases energy. At low intercalant concentrations, $\Delta E_{int}$ is again more negative for 0° stacking, but somewhere between 25% and 50% intercalations, the trend reverses and $\Delta E_{int}$ is lower for 180° stacking. We speculate that the change in the favorable intercalation site might be due to a crossing of the percolation threshold around 33%, at which point the in-plane interaction of intercalants becomes significant and changes the energetics of the favorable intercalation site. (See Supplementary Figs. 14, and 15 for additional DFT calculations including bulk structures and alternative intercalation sites.)

Based on the equilibrium structures predicted by DFT, we make a tentative picture of what is transpiring in the hPLD- and CVT-grown samples. At low $x$, a small amount of intercalation will switch the stable

stacking orientation from 180° to 0°. We can reach both sides of this phase transition in our bulk crystals, where a pristine sample with $x = 0$ shows homogeneous 180° stacking (Fig. 5a), and an intercalated sample with $x \sim 0.11$ shows homogeneous 0° stacking (Fig. 5b). Then at higher $x$, Nb$_{1+x}$Se$_2$ becomes thermodynamically more stable with 180° stacking again. Within the parameter space of our growth conditions, we only reached the boundary of the second transition around $x \sim 0.29$ in the thin films, where we observed both 0° and 180°-stacked layers coexisting on the nanoscale. Here, the intercalants are not homogeneously distributed but preferentially reside in the 180°-stacked layers, while leaving the 0°-stacked layers much emptier in comparison. The stacking selectivity of the intercalants points to a phase separation mechanism, perhaps spinodal decomposition, where a homogeneously intercalated $x \sim 0.29$ phase is unstable against nanoscale segregation into 180°-stacked layers with local $x > 0.29$ and 0°-stacked layers with local $x < 0.29$ (Fig. 6c). Our CVT-grown crystals could not reach the highly intercalated, 180°-stacked phase, even at a local level probed by STEM (Fig. 6d). When we synthesized crystals with a target composition as high as $x = 0.35$, we only stabilized a layered Nb$_{1+x}$Se$_2$ phase with $x \sim 0.20$. The thin films must have some specific kinetic pathways that enable them to realize a high degree of self-intercalation with 180° stacking, such as the high kinetic energy of laser-ablated Nb adatoms, or smaller film domains on the order of ~100 nm with plenty of edges and boundaries (see Supplementary Fig. 16 for AFM images), which better facilitate atomic motion and rearrangements. The thin films are also more prone to a loss of Se in the hPLD process, which could provide an alternative route towards self-intercalation and stacking change (removing Se rather than inserting Nb)[51]. Further experimental work is needed to pin down the exact microscopic mechanisms involved.

In conclusion, we have used STEM to directly visualize a stacking-selective self-intercalation phenomenon in epitaxial Nb$_{1+x}$Se$_2$ films. Presently, the highly intercalated 180°-stacked layers and sparsely intercalated 0°-stacked layers are interspersed on a nanometer length scale, such that bulk measurements of the entire film do not reveal any signatures of superconductivity. The importance of kinetic pathways

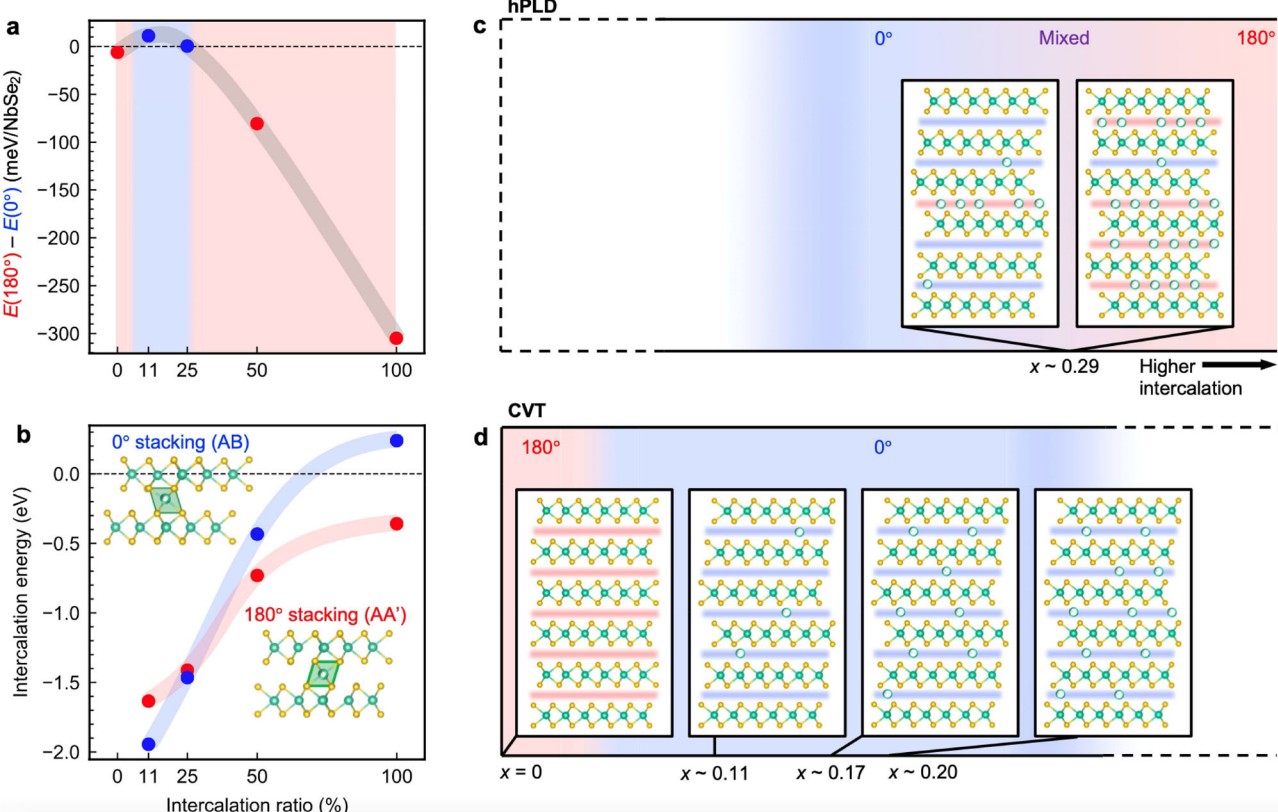

**Fig. 6 | Proposed microscopic picture of stacking-selective self-intercalation. a** The difference in DFT total energies per formula unit between 180°- and 0°-stacked bilayer $Nb_{1+x}Se_2$ with varying concentrations of Nb intercalants at the interstitial octahedral sites. A negative (positive) value of $E(180°) - E(0°)$ implies that the 180° (0°) structure is more stable and is indicated by the red (blue) color of the data points and shaded regions. **b** DFT-computed change in total energy upon intercalation into the octahedral voids of 180°- and 0°-stacked bilayer $Nb_{1+x}Se_2$, as shown in the insets. **c, d** Proposed phase diagrams for $Nb_{1+x}Se_2$ grown via hPLD and chemical vapor transport (CVT), respectively. The hPLD-grown films lie at the phase boundary between 0° and 180° stacking, whereas our CVT-grown crystals have not reached the second 180°-stacked phase.

suggests that further optimization could be possible. A future goal would be to improve the epitaxy, such that these two phases can be deterministically separated to some extent on an appropriate length scale to realize regions of metallicity and regions of superconductivity. Such films could then be harnessed for applications involving atomically-sharp superconductor-metal junctions, or devices where $Nb_{1+x}Se_2$ can serve as a metallic electrode for superconducting $NbSe_2$. Alternatively, after film deposition, electrochemical intercalation with a second species could be applied, such that the 180°-stacked layers host mostly Nb intercalants, while the 0°-stacked layers receive more of the new species and adopt distinct properties.

## Methods

### Film growth and parameters

$Nb_{1+x}Se_2$ films were grown on *c*-cut sapphire (aluminum oxide, $Al_2O_3$) substrates via hybrid pulsed laser deposition (hPLD) in ultrahigh vacuum[44,52]. A pulsed KrF excimer laser (Coherent LPX PRO 210 F) was utilized to ablate pure niobium (99.9%) targets at a repetition rate of 1−5 Hz. Pure selenium (99.999%) was evaporated from a Knudsen cell at a flux of roughly 2 Å/s, which was measured with a quartz crystal microbalance. The Nb:Se flux ratio was roughly 1:1800. The depositions were carried out at different temperatures for a few hours, followed by annealing both with and without selenium flux. After annealing, additional selenium could be deposited on the film at room temperature for capping, which protected the films. The detailed growth conditions and in-situ RHEED monitoring of the films are listed in the supplemental information (see Supplementary Fig. 1 and Table 1).

### Bulk crystal growth

Single crystals of $Nb_{1+x}Se_2$ were grown by chemical vapor transport with iodine as the transporting agent. Powder samples were prepared by heating mixtures of Nb (99.9%) and Se (99.999%) in an evacuated silica tube at 800 °C for one day. The powder sample and iodine (5 mg/cm³) were sealed in a silica tube and heated in a two-zone furnace for 10 days with the temperature of the source and growth zones fixed at 1000 °C and 700 °C, respectively. The typical size of the crystals obtained was $2 × 2 × 0.2$ mm. Starting Nb:Se compositions of 1.08:2, 1.25:2, and 1:35:2 yielded actual crystals of $Nb_{1.11}Se_2$, $Nb_{1.17}Se_2$, and $Nb_{1.20}Se_2$, as determined by energy dispersive x-ray spectroscopy.

### XRD

XRD measurements were performed to confirm the interlayer distance of the $Nb_{1+x}Se_2$ films, as well as to determine the film thickness from Kiessig fringes. We used a Huber four-circle diffractometer equipped with a Cu $K_\alpha$ source and a Mythen 1D detector from Dectris.

### STM and atomic force microscopy (AFM)

STM measurements were performed using a customized room-temperature system from Unisoku. W tips were prepared by electro-chemical etching, followed by in-situ sharpening with electron beam heating. In order to obtain a pristine surface for constant-current imaging, a Se capping layer was deposited on the film, prior to removal of the sample from the hPLD chamber. The Se capping layer was then removed via heating inside the STM load lock to expose a fresh surface. AFM measurements were performed in an ambient environment using a Bruker system with the PeakForce Tapping mode. Different

surface morphologies can be observed for the $Nb_{1+x}Se_2$ films with different thicknesses.

## LEED, XPS, and ARPES

LEED, XPS, and ARPES measurements were performed in two systems. The first system includes a SPECS ErLEED 150 instrument and a Kratos AXIS Ultra spectrometer with a monochromatized Al $K_\alpha$ source, with which we obtained LEED (Supplementary Fig. 4a, b) and XPS data (Supplementary Figs. 9, 10). The second system is a NanoESCA (Scienta Omicron), with which we obtained both LEED (Supplementary Fig. 4c, d) and ARPES data (Fig. 4a, b, and Supplementary Fig. 4e, f). The ARPES data in Fig. 4a, b were acquired with a single shot as a cube ($k_x$-$k_y$-$E_B$) with an energy resolution of 50 meV using a non-monochromatized HeI UV-source. All LEED, XPS, and ARPES measurements were done at room temperature. To preserve the sample surface, films were capped with a Se layer, which was subsequently removed by heating at 250–300 °C for 10–30 min prior to measurements.

We fit the Se 3$d$ XPS core level spectra using the CasaXPS software[53]. The fitting functions were primarily composed of Gaussian-Lorentzian mixtures on top of a Shirley background. For the Nb and Se $3d_{3/2}$ and $3d_{5/2}$ core levels, we constrained the doublet ratio to be 2:3 and fixed the doublet splitting to literature values[54], while the FWHM was left as a free-fitting parameter. We calibrated the binding energies to the Fermi edge observed in a valence band spectrum of the film. To determine the band positions in the ARPES intensity cuts, we took the maxima from a combination of energy distribution curves and momentum distribution curves. Assuming that the photoelectrons in ARPES come from roughly three times the inelastic mean free path, which is approximately 1 nm for electrons with kinetic energies in the range of 15 eV[55], and an interlayer distance of roughly 0.65 nm, most of the signal comes from the top five layers, which covers the majority of the entire film thickness.

## Transport measurement

We performed transport measurements on 5 mm × 5 mm thin films with two methods and obtained similar results. In the first method, we removed the selenium capping and evaporated six gold electrodes onto the exposed $Nb_{1+x}Se_2$ film in a Hall bar geometry. We used a wire bonder to connect gold wires with a diameter of 50 μm between the film electrodes and the pads on the puck. In the second method, we mechanically removed portions of the $Nb_{1+x}Se_2$ film to define a Hall bar geometry. The film remained capped with a Se layer, so we attached gold wires by scratching away the Se capping at contact points and applying silver epoxy. We collected the resistivity and magnetoresistance data using the resistivity option of a Physical Property Measurement System (PPMS) from Quantum Design. The measurements were carried out in the He-3 insert of the PPMS, with a base temperature of 300 mK and a maximum magnetic field of 14 T. We symmetrized the magnetoresistance data with respect to the field direction.

## Scanning transmission electron microscopy

STEM specimens in a cross-sectional orientation were prepared by a focused ion beam technique. Prior to the ion cutting and milling process, the surface of the Se-capped $Nb_{1+x}Se_2$ thin film was coated with electron beam-deposited Pt and ion beam-deposited Pt to protect the material from ion beam damage[56].

STEM studies were conducted using a spherical aberration-corrected STEM (JEM-ARM200F, JEOL Co. Ltd.) equipped with a cold field-emission gun and a DCOR probe Cs-corrector (CEOS GmbH) operated at 200 kV. The STEM images were obtained by an annular dark-field (ADF) detector with a convergent semi-angle of 20.4 mrad and collection semi-angles of 70–300 mrad. In order to make precise measurements of lattice constants, ten serial frames were acquired

with a short dwell time (2 μs/pixel), aligned, and added afterward to improve the signal-to-noise ratio (SNR) and to minimize the image distortion of HAADF images. EELS acquisition was performed by a Gatan GIF Quantum ERS imaging filter equipped with a Gatan K2 Summit camera with a convergent semi-angle of 20.4 mrad and a collection semi-angle of 111 mrad. EELS 2D spectrum imaging was performed with a dispersion of 0.5 eV/channel and 1436 eV drift-tube energy with a 4000-pixel wide detector for the simultaneous acquisition of signals of Se-$L_{2,3}$, Al-K, and Nb-$L_{2,3}$ edges. The EELS line scan was performed with a dispersion of 0.1 eV/channel and 45 eV drift tube energy with a 4000-pixel wide detector for the simultaneous acquisition of signals of Nb-$N_{2,3}$, Nb-$M_{2,3}$, Nb-$M_{4,5}$ and Se-$M_{4,5}$ edges. The raw spectrum image data were denoised by applying a principal component analysis (PCA) with the multivariate statistical analysis (MSA) plugin (HREM Research Inc.) in Gatan DigitalMicrograph and then smoothed using a spatial filter in Gatan DigitalMicrograph. The first and second derivative analyses of EELS spectra were conducted using a built-in function in Gatan DigitalMicrograph. HAADF-STEM images of $2H_a$-phase $NbSe_2$ were simulated using the abTEM code[57]. Atomically resolved HAADF-STEM images have been quantified, where the coordinate positions of Nb and Se ions were determined by a combination of center-of-mass and 2D Gaussian fitting in high-resolution HAADF images, and then used to calculate the interlayer spacing between adjacent $NbSe_2$ layers and Se-Nb-Se angles in $NbSe_2$ layers[58,59].

## Density functional theory

To evaluate the phase stability and electronic structure of the pristine and intercalated $NbSe_2$, DFT calculations were performed using the Vienna Ab initio Simulation Package (VASP) based on the projected augmented wave method[60,61]. The electron exchange-correlation function was described by the Perdew-Burke-Ernzerhof (PBE) parametrization of the generalized gradient approximation (GGA). The wave functions were expanded on a plane-wave basis with a cutoff energy of 520 eV using a k-mesh density of 50 Å$^{-1}$. The tolerance of total energy convergence for self-consistent field calculations was set to be $10^{-5}$ eV. All structures were fully optimized until the maximal Hellmann–Feynman forces on atoms were less than 0.01 eV/Å. The Becke88 optimization (optB88) was used to accurately account for non-local vdW forces[62].

## Data availability

Relevant data supporting the key findings of this study are available within the article and the Supplementary Information file. All raw data generated during the current study are available from the corresponding authors upon request.

## Code availability

All the codes used for this study are available from the corresponding authors upon request.

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

## Acknowledgements

This project has received funding from the European Union's Horizon 2020 research and innovation program under Grant Agreement No. 823717 – ESTEEM3. The authors thank the Max Planck Society for financial support. The authors gratefully acknowledge the insightful discussions with T.T.M. Palstra, W. Sigle, U. Wedig, and A. Yaresko, TEM support by K. Hahn, hPLD support by B. Stuhlhofer and G. Cristiani, and technical support by K. Pflaum and M. Dueller. The Lichtenberg high-performance computer of the TU Darmstadt is gratefully acknowledged for the computational resources where the calculations were conducted for this project.

## Author contributions

H.W. and D.H. conceived the project. H.W., C.Y., and P.A.v.A. conducted the STEM measurements and related data analysis. J.Z. and D.H. grew the NbSe$_2$ thin films and performed XRD, transport, AFM, and STM measurements. M.I. grew single crystals via chemical vapor transport. K.K. and U.S. carried out the LEED, XPS, and ARPES measurements. C.S. and H.B.Z. performed DFT calculations and related data analysis. J.D. prepared TEM samples and conducted SEM-EDS analysis. H.B.Z., P.A.v.A, and D.H. supervised this work, and H.T. provided insights and interpretation of the results. All authors contributed to the work and commented on the paper.

## Funding

## Competing interests

The authors declare no competing interests.
