## [Peer Review File · Nature Communications]

Direct visualization of stacking-selective self-intercalation in epitaxial $\text{Nb}_{1+x}\text{Se}_2$ filmsEditorial Note: Parts of this Peer Review File have been redacted as indicated to remove third-party material where no permission to publish could be obtained.

REVIEWER COMMENTS

Reviewer #1 (Remarks to the Author):

The manuscript by Hongguang Wang et al. reports a detailed atomic-resolution STEM investigation, combined with theoretical calculations, on the study of the self-intercalation mechanism in Nb-rich niobium selenide epitaxial films grown by PLD on sapphire substrates. Authors visualize a stacking-selective self-intercalation process where excess Nb intercalants drive the formation of 3R-AB stacking instead of AA' stacking, in contradiction to previous reports. After reading the manuscript, I see that authors intend to present this mechanistic process as a universal phenomenon, which I don't agree with and I think authors do not provide enough evidence to support that claim, therefore I cannot recommend the paper for publication in Nature Communications in the current state. However, I still think the findings are sound and it may be interesting for the community. In the following I list of some questions to be addressed.

1) As a general comment, I find very disturbing the notation the authors use referring to 0° and 180° stacking instead of the conventional 2H-AA' and 3R-AB stacking. I suggest the authors to stick to more widely accepted terminology instead of creating new unnecessary notations. Personally, I found the draft difficult to follow at some points due to this matter.

2) It is known from the literature that higher chemical potential of the metal during the growth process of TMDs favors the 3R stacking [Phys. Rev. B 93, 041420 (2016); Sci Rep 8, 2143 (2018); Adv. Mater. 2018, 30, 1704674; Nat Commun 9, 199 (2018)]. Authors indirectly discuss this in the particular case of Nb-Se system in line 56, saying that $x > 0.07$ induce a transition from AA' to AB stacking, citing references [24-27]. However, generally, these studies were made on samples grown by chemical methods such as CVD or CVT, methods closer to equilibrium that allow more thermodynamically stable phases to nucleate. In the present work, samples are epitaxially grown by PLD, which allows layer-to-layer control and the possibility of stabilizing metastable phases and processes in non-equilibrium conditions. One wonders if the discrepancies authors try to debate between their findings on the origin of the stacking switching and the previously reported are simply due to the use of different growth methods, which allows new growth kinetics to happen and processes not observed before. The growth method is a key factor in controlling and realizing materials.

Thanks to more and more microscopy works, as the present one, we are finding out that most of the discrepancies found in literature in the last years regarding the intercalation process, polytyping, seems to have the origin on the growth method employed. I think this issue should be properly addressed and discussed. In my opinion, this doesn't invalidate their findings but the story of the paper, conclusions and abstract may change considerably. I think the author's discoveries may not necessarily invalidate the previous findings, and I think there is not enough compelling arguments to present these observations as a general mechanism. I invite the authors to review the manuscript in this regard and avoid generalization if they cannot provide an unambiguous comparison that supports their arguments.

3) Continuing with the growth conditions, I see the growth of the samples require temperatures as high as 600° C sometimes. The chalcogen Se is problematic due to its low evaporation temperature. Some experiments in V-Se samples showed that post-growth annealing at temperatures above 400 °C results in a significant loss of Se [Adv Mater Interfaces 2020, 7 (15), 1–9; Adv Mater 2019, 31 (40), 1903779], inducing phase transitions or emerging ferromagnetism. This is important. Se evaporation will imbalance the chemical potential (previous comment), thus Se vacancies may be another parameter to consider for the conclusions of the paper and the role of the Nb intercalations. Have authors consider the possibility of Se vacancies during the growth or annealing process? How to make sure no Se was evaporated? How may change the formation energies of different stacking types?

4) On the other hand, I found interesting Figure S2. If I understood correctly from the main text, the authors attribute the bright atoms in the topographic STM image to the Se surface, and then the dark and bright contrast variations to Nb intercalations? In my understanding, in a topographic image, adatoms should show brighter than the Se surface and, if vacancies (Se here), as dark atoms. How can we understand this image? This means there are both adatoms and Se vacancies? This question should be addressed; I am not sure what message is the image delivering.

5) The STEM shows clear interlayer expansion when the vdW gap is filled with intercalations. I found this hard to understand. Intercalants will create covalent bonds [Nature 581, 171-177 (2020)], therefore the large weakly bonded vdW gap should be eliminated or partially eliminated. For instance, as an example from a sister family system, the c axis of VSe₂ (1:2, vdW gap fully empty) and VSe (1:1, vdW gap fully occupied) is 0.613 and 0.578 nm, respectively (~ - 6%). How can we understand the lattice expansion in the case of Nb-Se?

6) The staggering of intercalants is quite interesting. Here it is only shown for the case of 7–8-layer films. Is the same for the rest of samples? What about the thicker films? Seems to me this staggering mechanism may be very well connected with the PLD growth process, as discussed in comment 2. The formation of intercalated-superstructures and twisting has been also reported for CVD-grown Cr-Te nanoplates [Nano Lett. 2021, 21, 9517–9525].

Reviewer #2 (Remarks to the Author):

The authors present a thorough investigation of Nb_{1+x}Se₂ synthesized by PLD. They go beyond prior observations and demonstrations by correlating the self-intercalation with layer stacking, unambiguously identifying the nature and position of the intercalant, observing ARPES spectra that appears to be a mixture of 2H_a and 2H_b, reported on the electronic properties. This work is well written and free of errors and unjustified conclusions. I recommend publication as is.

Minor comment:

The authors state the photon energy of their monochromated Al K α source as 1486.6eV. While this is commonly reported in the literature, it is incorrect and likely a legacy of un-monochromated dual anode sources where the weighted average of the k-alpha1 (1486.7eV) and the k-alpha2 (1486.3eV) was used. Since a monochromator will be optimized to the maximum intensity the result will be a k-alpha1 line at 1486.7eV. While the incorrect 1486.6 eV is quite prevalent throughout the literature (and even some vendor websites), I don't think this error should be propagated to a nature journal. Unless the authors optimized their monochromator away from the maximum intensity then the photon energy would be 1486.7 eV.

The authors can see the work by J. Schweppe R.D. Deslattes T. Mooney and C.J. Powell as well as the LBL tables which both confirm the energy of the Ka1 line

[https://doi.org/10.1016/0368-2048\(93\)02059-U](https://doi.org/10.1016/0368-2048(93)02059-U)

https://xdb.lbl.gov/Section1/Table_1-2.pdf

Optional suggestion:

When reviewing the SI, I noted that the results and interpretation of the XPS and XRD data is similar and consistent with those reported recently by Litwin et al (for Nb $_{1+x}$ Se $_2$)[1] and Bonilla et al (for VSe $_2$)[2] and as such may warrant comparison. Since Bonilla et al also reported on the surface structure observed with STM it would be interesting to consider whether all observations are consistent.

<https://doi.org/10.1116/6.0002593>

<https://doi.org/10.1002/admi.202000497>

Reviewer #3 (Remarks to the Author):

****THIS REPORT WAS WRITTEN IN COLLABORATION WITH REFEREE #4**

H Wang et al study the intercalation of excess Nb within hybrid PLD grown Nb $_{1+x}$ Se $_2$ thin films, and they observe a strong interaction between the layer stacking order and intercalation. Using atomically resolved STEM imaging and EELS analysis, the authors find that their films contain both 180° and 0° stacking, and that intercalants preferentially occupy the octahedral sites between 180° stacked layers. Negligible intercalation was found between 0° stacked layers. The experimental data suggest that Nb intercalants energetically prefer the octahedral 180° sites, and this is confirmed via DFT calculations. The authors also argue that the presence of 0° stacking is driven by the inter-layer repulsion of Nb intercalants and a staging mechanism. The effects of intercalation on the electronic properties are investigated with ARPES and resistivity measurements. Both the intercalation of TMDs and the layer stacking in TMDs are topics of interest, and this manuscript presents an interesting system where intercalation and layer stacking cooperate in a non-trivial manner. The manuscript is well-written, and the characterization and analysis are complete. Below are some questions / comments for the authors.

1. The term staging is most often associated with Li intercalation into graphite, which progresses from

stage 4, to stage 3, stage 2, and finally stage 1, which corresponds to fully intercalated graphite. Do the authors propose that similar stages occur for Nb_{1+x}Se₂? Have different stages been observed experimentally? The authors should clarify their use of the term staging.

2. The authors propose that the 0° layer stacking is driven by the inter-layer repulsion of Nb intercalation. So the 0° stacking forms to separate layers of fully intercalated 180° stacking. If this argument is accurate, then 0° stacking should only form in close proximity to intercalated 180° stacking. But the data in Fig. 3 seems to contradict this argument. In Fig 3d, the NbSe₂ film is entirely 0° stacked, except for a single layer of intercalated 180° stacking at the substrate interface. The topmost 0° stacked layers cannot be driven by an inter-layer Nb repulsion mechanism, since there is no Nb intercalation nearby. A similar situation is found in Fig. 3c.

3. Have the authors grown hPLD thin films of stoichiometric NbSe₂ (with no self-intercalation) and studied the layer stacking? Is it possible that the hPLD growth conditions yield mixed layer stacking even in the absence of self-intercalation?

4. What is the lateral length-scale of the different layer stacking orders and intercalated layers? Do the authors have any wide field-of-view images showing changes in the layer stacking and intercalant structure? For instance, a lateral transition from 0° stacking to 180° stacking? Alternatively, do the authors have any selected area electron diffraction of the FIB'd samples, which would provide information on the layer stacking averaged across a large region?

5. For the ARPES measurements, what is the penetration depth? Is the entire film thickness being probed?

6. The text references a yellow arrow in Fig. 2f, but the arrow is pink. Also, the text refers to an orange arrow in Fig. 2i, but the arrow looks green.

7. For the Nb oxidation state analysis, did the authors look at the fine structure of the Nb L-edge (Fig. 2b) to confirm their findings at the Nb N-edge? Does the N-edge offer advantages for chemical analysis compared to the L-edge?

8. In Fig. 1g, how were the error bars calculated?

Reviewer #4 (Remarks to the Author):

**THIS REPORT WAS WRITTEN IN COLLABORATION WITH REFEREE #3

H Wang et al study the intercalation of excess Nb within hybrid PLD grown Nb_{1+x}Se₂ thin films, and they observe a strong interaction between the layer stacking order and intercalation. Using atomically resolved STEM imaging and EELS analysis, the authors find that their films contain both 180° and 0°

stacking, and that intercalants preferentially occupy the octahedral sites between 180° stacked layers. Negligible intercalation was found between 0° stacked layers. The experimental data suggest that Nb intercalants energetically prefer the octahedral 180° sites, and this is confirmed via DFT calculations. The authors also argue that the presence of 0° stacking is driven by the inter-layer repulsion of Nb intercalants and a staging mechanism. The effects of intercalation on the electronic properties are investigated with ARPES and resistivity measurements. Both the intercalation of TMDs and the layer stacking in TMDs are topics of interest, and this manuscript presents an interesting system where intercalation and layer stacking cooperate in a non-trivial manner. The manuscript is well-written, and the characterization and analysis are complete. Below are some questions / comments for the authors.

1. The term staging is most often associated with Li intercalation into graphite, which progresses from stage 4, to stage 3, stage 2, and finally stage 1, which corresponds to fully intercalated graphite. Do the authors propose that similar stages occur for $\text{Nb}_{1+x}\text{Se}_2$? Have different stages been observed experimentally? The authors should clarify their use of the term staging.
2. The authors propose that the 0° layer stacking is driven by the inter-layer repulsion of Nb intercalation. So the 0° stacking forms to separate layers of fully intercalated 180° stacking. If this argument is accurate, then 0° stacking should only form in close proximity to intercalated 180° stacking. But the data in Fig. 3 seems to contradict this picture. In Fig 3d, the NbSe_2 film is entirely 0° stacked, except for a single layer of intercalated 180° stacking at the substrate interface. The topmost 0° stacked layers cannot be driven by an inter-layer Nb repulsion mechanism, since there is no Nb intercalation nearby. A similar situation is found in Fig. 3c. Can the authors comment on Fig 3 and the proposed staging mechanism?
3. Have the authors grown hPLD thin films of stoichiometric NbSe_2 (with no self-intercalation) and studied the layer stacking? Is it possible that the hPLD growth conditions yield mixed layer stacking even in the absence of self-intercalation?
4. What is the lateral length-scale of the different layer stacking orders and intercalated layers? Do the authors have any wide field-of-view images showing changes in the layer stacking and intercalant structure? For instance, a lateral transition from 0° stacking to 180° stacking? Alternatively, do the authors have any selected area electron diffraction of the FIB'd samples, which would provide information on the layer stacking averaged across a large region?
5. For the ARPES measurements, what is the penetration depth? Is the entire film thickness being probed?
6. The text references a yellow arrow in Fig. 2f, but the arrow looks pink. Also, the text refers to an orange arrow in Fig. 2i, but the arrow looks green.
7. For the Nb oxidation state analysis, did the authors look at the fine structure of the Nb L-edge (Fig. 2b) to confirm their findings at the Nb N-edge? Does the N-edge offer advantages for chemical analysis compared to the L-edge?
8. In Fig. 1g, how were the error bars calculated?

Response to the Reviewers and Revisions

Reviewer #1:

The manuscript by Hongguang Wang et al. reports a detailed atomic-resolution STEM investigation, combined with theoretical calculations, on the study of the self-intercalation mechanism in Nb-rich niobium selenide epitaxial films grown by PLD on sapphire substrates. Authors visualize a stacking-selective self-intercalation process where excess Nb intercalants drive the formation of 3R-AB stacking instead of AA' stacking, in contradiction to previous reports. After reading the manuscript, I see that authors intend to present this mechanistic process as a universal phenomenon, which I don't agree with and I think authors do not provide enough evidence to support that claim, therefore I cannot recommend the paper for publication in Nature Communications in the current state. However, I still think the findings are sound and it may be interesting for the community. In the following I list of some questions to be addressed.

Our response:

We thank the reviewer for their interest and thoughtful evaluation of our work. We agree that kinetics can play as much a role as thermodynamics, and that our observations in $\text{Nb}_{1+x}\text{Se}_2$ thin films grown by our hPLD technique may not be universal to $\text{Nb}_{1+x}\text{Se}_2$ prepared in other forms, such as bulk crystals. To clarify this point, we have performed comparative STEM experiments on a series of bulk $\text{Nb}_{1+x}\text{Se}_2$ single crystals grown by chemical vapor transport (CVT). Please refer to our response to Comment 2 for a detailed explanation.

Comment 1:

1) As a general comment, I find very disturbing the notation the authors use referring to 0° and 180° stacking instead of the conventional 2H-AA' and 3R-AB stacking. I suggest the authors to stick to more widely accepted terminology instead of creating new unnecessary notations. Personally, I found the draft difficult to follow at some points due to this matter.

Our response:

We regret any confusion caused by our notation. The reason we did not simply apply the terms 2H-AA' and 3R-AB is that we actually observed a total of four stacking polytypes: $2H_a$ -AA' (180° stacking), $2H_b$ -AB (0° stacking), 3R-ABC (0° stacking), and $4H_x$ -ABAC (0° stacking). In fact, in the thin films, $2H_b$ and $4H_x$ were more prevalent than 3R. Instead of lumping the $2H_b$ and $4H_x$ polytypes into the 3R polytype, we decided to refer to these three simply as 0° -stacked polytypes, and the $2H_a$ polytype as a 180° -stacked polytype. Our intention was to make our description accessible to a more general audience who might not be familiar with the Ramsdell notation.

Actions taken:

We added an additional sentence in the subsection "Stacking-selective self-intercalation" to try to better clarify our notation (Page 8 in the main manuscript):

"To summarize, we observed a total of four polytypes: $2H_a$ -AA', which we refer to as a 180° -stacked polytype, and $2H_b$ -AB, 3R-ABC, and $4H_x$ -ABAC, which we collectively refer to as 0° -stacked polytypes."

Comment 2:

It is known from the literature that higher chemical potential of the metal during the growth process of TMDs favors the 3R stacking [Phys. Rev. B 93, 041420 (2016); Sci Rep 8, 2143 (2018); Adv. Mater. 2018, 30, 1704674; Nat Commun 9, 199 (2018)]. Authors indirectly discuss this in the particular case of Nb-Se system in line 56, saying that $x > 0.07$ induce a transition from AA' to AB stacking, citing references [24-27]. However, generally, these studies were made on samples grown by chemical methods such as CVD or CVT, methods closer to equilibrium that allow more thermodynamically stable phases to nucleate. In the present work, samples are epitaxially grown by PLD, which allows layer-to-layer control and the possibility of stabilizing metastable phases and processes in non-equilibrium conditions. One wonders if the discrepancies authors try to debate between their findings on the origin of the stacking switching and the previously reported are simply due to the use of different growth methods, which allows new growth kinetics to happen and processes not observed before. The growth method is a key factor in controlling and realizing materials.

Thanks to more and more microscopy works, as the present one, we are finding out that most of the discrepancies found in literature in the last years regarding the intercalation process, polytyping, seems to have the origin on the growth method employed. I think this issue should be properly addressed and discussed. In my opinion, this doesn't invalidate their findings but the story of the paper, conclusions and abstract may change considerably. I think the author's discoveries may not

necessarily invalidate the previous findings, and I think there is not enough compelling arguments to present these observations as a general mechanism. I invite the authors to review the manuscript in this regard and avoid generalization if they cannot provide an unambiguous comparison that supports their arguments.

Our response:

We thank and agree with the reviewer on this statement. Our STEM images disclose that epitaxial films of $\text{Nb}_{1+x}\text{Se}_2$ near a global $x \sim 0.29$ show a nanoscale coexistence of 180° -stacked layers with a large number of Nb intercalants, perhaps several tens of percentage occupancy (considering the average x value), and 0° -stacked layers whose number of Nb intercalants falls close to the detection limit. To evaluate the role kinetics and growth conditions play in these observations, and how they relate (if at all) to previous bulk reports, we performed a new comparative study on single crystals of $\text{Nb}_{1+x}\text{Se}_2$ grown by chemical vapor transport (CVT).

Figure R1 presents a series of HAADF-STEM images for four $\text{Nb}_{1+x}\text{Se}_2$ crystals with increasing average values of x . At $x = 0$, NbSe_2 crystallizes in the $2H_a$ polytype with 180° stacking, and the vdW gaps are clear of intercalants [Fig. R1(a)]. At $x \sim 0.11$, the NbSe_2 layers uniformly adopt the $3R$ configuration with 0° stacking [Fig. R1(b)]. There are faint signatures of some intercalants inside the octahedral voids of the vdW gaps, perhaps to a degree that is qualitatively similar to what we observe between the 0° -stacked layers of our thin films. As we continued to synthesize crystals with increasing x via CVT, the degree of self-intercalation began to saturate. An initial Nb:Se stoichiometry of 1.25:2 resulted in $\text{Nb}_{1+x}\text{Se}_2$ crystals with average $x \sim 0.17$, showing again a homogeneous 0° -stacked phase, but with more intercalants inside their octahedral voids [Fig. R1(c)]. A greater starting Nb:Se stoichiometry of 1.35:2 resulted in $\text{Nb}_{1+x}\text{Se}_2$ crystals with average $x \sim 0.20$. These crystals contained domains of additional Nb-Se phases (see Supplementary Fig. S13); nevertheless, in the regions with $1H$ layers of NbSe_2 , we still observed homogeneous 0° stacking [Fig. R1(d)]. The degree of intercalation between the 0° -stacked layers here seems to be much greater than the 0° -stacked layers of our thin films [e.g., Figs. 1(g) or 3(c) of the main text], but still less than the 180° -stacked layers of our thin films [e.g., Fig. 3(a) of the main text]. Contrary to our previous suggestion, we do not invalidate the previous bulk studies at low x , but reproduce them with STEM. It appears that x is a crucial parameter. However, with the CVT crystals, we could not reach the higher x values comparable to our thin films, at which highly intercalated, 180° -stacked layers begin to appear locally.

Figure R1. Self-intercalation in single-crystal $\text{Nb}_{1+x}\text{Se}_2$ grown via chemical vapor transport. (a)–(d) HAADF-STEM images of the cross-section of bulk $\text{Nb}_{1+x}\text{Se}_2$ with nominal $x = 0, 0.11, 0.17,$ and 0.20 , respectively. A close-up of the region enclosed by the yellow box is shown for each panel.

To reconcile what we observe in $\text{Nb}_{1+x}\text{Se}_2$ grown via hPLD and CVT in different global x regimes, we performed new DFT calculations over a greater range of x values. We constructed supercells of bilayer $\text{Nb}_{1+x}\text{Se}_2$ with 180° (AA') and 0° (AB) stackings with Nb intercalants inside the octahedral voids. The supercells had chemical formulae Nb_2Se_4 , $\text{Nb}_{19}\text{Se}_{36}$, $\text{Nb}_9\text{Se}_{16}$, $\text{Nb}_{10}\text{Se}_{16}$, and Nb_3Se_4 , which correspond to an occupancy of the interstitial octahedral sites of 0%, 11%, 25%, 50%, and 100%. (For bilayer $\text{Nb}_{1+x}\text{Se}_2$ surrounded by vacuum, there is only one octahedral void per unit cell, so the occupancy of octahedral voids is given by $2x$, in contrast to bulk $\text{Nb}_{1+x}\text{Se}_2$, where the occupancy is given by x .) At 0% intercalation, the total energy after structural relaxation of 180° -stacked bilayer NbSe_2 is only 2 meV per formula unit smaller than that of 0° -stacked bilayer NbSe_2 [Fig. R2(a)]. The near degeneracy of these two stackings reflects the weak interlayer coupling of vdW

nature in NbSe₂, but the prediction of a slightly more energetically favorable 180° stacking is in line with the observed 2H_a polytype in bulk, pristine NbSe₂. At 11% and 25% intercalations, the 0°-stacked bilayer is actually lower in energy than the 180°-stacked bilayer, but the situation is quickly reversed at 50% and 100% intercalations, where the 180°-stacked bilayer is again more stable. DFT predicts two polytype transformations with increasing x .

Figure R2. Proposed microscopic picture of stacking-selective self-intercalation. (a) The difference in DFT total energies per formula unit between 180°- and 0°-stacked bilayer Nb_{1+x}Se₂ with a varying concentration of Nb intercalants at the interstitial octahedral sites. A negative (positive) value of $E(180^\circ) - E(0^\circ)$ implies that the 180° (0°) structure is more stable and is indicated by the red (blue) color of the data points and shaded regions. (b) DFT-computed change in total energy upon intercalation into the octahedral voids of 180°- and 0°-stacked bilayer Nb_{1+x}Se₂. (c) and (d) Proposed phase diagram for Nb_{1+x}Se₂ grown via hPLD and CVT, respectively. The hPLD-grown films lie at the phase boundary between 0° and 180° stacking, whereas our CVT-grown crystals have not reached the second 180°-stacked phase.

We also computed the intercalation energy in Fig. R2(b), defined as $\Delta E_{\text{int}} = [E(\text{Nb}_{1+x}\text{Se}_2) - E(\text{NbSe}_2) - xE(\text{Nb})]/x$, where $E(\text{Nb}_{1+x}\text{Se}_2)$ and $E(\text{NbSe}_2)$ are the DFT total energies after structural relaxation of the intercalated and pristine structures and $E(\text{Nb})$ is the DFT-computed energy required to take one Nb atom from bulk Nb. In all but one case, ΔE_{int} is negative, meaning that self-intercalation actually releases energy. Again, at low intercalant concentrations, ΔE_{int} is more negative for 0° stacking, but somewhere between 25% and 50% intercalation, the trend reverses and ΔE_{int} is lower for 180° stacking. We speculate that the change in the favorable intercalation site might be due to a crossing of the percolation threshold around 33%, at which point the in-plane interaction of intercalants becomes significant and changes the energetics of the favorable intercalation site. (Supplementary Fig. S14, S15 for additional DFT calculations including bulk structures and alternative intercalation sites.)

Based on the equilibrium structures predicted by DFT, we make a tentative picture of what is transpiring in the hPLD- and CVT-grown samples. At low x , a small amount of intercalation will switch the stable stacking orientation from 180° to 0°. We can reach both sides of this phase transition in our bulk crystals, where a pristine sample with $x = 0$ shows homogeneous 180° stacking [Fig. R1(a)], and an intercalated sample with $x \sim 0.11$ shows homogeneous 0° stacking [Fig. R1(b)]. Then at higher x , Nb_{1+x}Se₂ becomes thermodynamically more stable with 180° stacking again. Within the parameter space of our growth conditions, we only observed this second transition in the thin films with $x \sim 0.29$, but as a nanoscale phase coexistence of highly intercalated, 180°-stacked layers and sparsely intercalated, 0°-stacked layers [Fig. R2(c)]. Our CVT-grown crystals could not reach the highly intercalated, 180°-stacked phase, even at a local level probed by STEM [Fig. R2(d)]. When we synthesized crystals with a target composition as high as $x = 0.35$, we only stabilized a layered Nb_{1+x}Se₂ phase with $x \sim 0.20$. The thin films must have some specific kinetic pathways that enable them to realize a high degree of self-intercalation with 180° stacking, such as the high kinetic energy of laser-ablated Nb adatoms, or smaller film domains on the order of ~100 nm with plenty of edges and boundaries (see Supplementary Fig. S16 for AFM images), which better facilitate atomic motion and rearrangements. The thin films are also more prone to a loss of Se in the hPLD process, which

could provide an alternative route towards self-intercalation and stacking change (removing Se rather than inserting Nb, as suggested by the reviewer). Further experimental work is needed to pin down the exact microscopic mechanisms involved.

Actions taken:

Based on the new STEM experiments and DFT calculations, we have substantially revised the story of our manuscript. Figures R1 and R2 are now included in the main manuscript as Figs. 5 (Page 11) and 6 (Page 12). Sections of our abstract (Page 1) and introduction (Page 3) have been revised, as well as the Discussion section (Pages 10-13).

Comment 3:

Continuing with the growth conditions, I see the growth of the samples require temperatures as high as 600° C sometimes. The chalcogen Se is problematic due to its low evaporation temperature. Some experiments in V-Se samples showed that post-growth annealing at temperatures above 400 °C results in a significant loss of Se [Adv Mater Interfaces 2020, 7 (15), 1–9; Adv Mater 2019, 31 (40), 1903779], inducing phase transitions or emerging ferromagnetism. This is important. Se evaporation will imbalance the chemical potential (previous comment), thus Se vacancies may be another parameter to consider for the conclusions of the paper and the role of the Nb intercalations. Have authors consider the possibility of Se vacancies during the growth or annealing process? How to make sure no Se was evaporated? How may change the formation energies of different stacking types?

Our response:

The reviewer raises an important point. It could be that the loss of Se at high temperatures (equivalent to an increase of the effective Nb:Se ratio) provides another kinetic pathway by which self-intercalation transpires in thin-film $\text{Nb}_{1+x}\text{Se}_2$. We do not currently have any experimental handle on what happens during the intermediate stages of the thin-film deposition process. We can say experimentally, though, that in the final state of our thin films after growth, we have not observed an abundance of Se vacancies. We have attempted to quantify the amount of Se vacancies seen at the surface of our thin films via STM imaging. In a 10×5 nm field of view, four Se vacancies can be observed out of roughly 561 Se sites, giving a vacancy ratio $< 0.8\%$ (Fig. R3). In terms of computing the formation energies of different stacking types, we think it is valid to omit the influence of 0.8% Se vacancies. In terms of reaction pathways by which self-intercalation actually occurs, however, the loss of Se could play an important role.

Figure R3. Se vacancies on the surface observed in an STM image. Four vacancies are identified on the Se surface. Scale bar: 2 nm.

Actions taken:

We have mentioned the loss of Se as a possible mechanism of self-intercalation in the Discussion section (Page 13):

“The thin films are also more prone to a loss of Se in the hPLD process, which could provide an alternative pathway towards self-intercalation and stacking change (removing Se rather than inserting Nb)⁵¹.”

Comment 4:

On the other hand, I found interesting Figure S2. If I understood correctly from the main text, the authors attribute the bright atoms in the topographic STM image to the Se surface, and then the dark and bright contrast variations to Nb intercalations? In my understanding, in a topographic image, adatoms should show brighter than the Se surface and, if vacancies (Se here), as dark atoms. How can we understand this image? This means there are both adatoms and Se vacancies? This question should be addressed; I am not sure what message is the image delivering.

Our response:

We thank the reviewer for their comment about the nature of the contrast shown in Fig. S2. In STM topographic images, a contrast may come from adatoms, vacancies, or an inhomogeneous background. As shown in Fig. R4(a), a vacancy is observed as an isolated dark region with no local maximum, surrounded by a hexagon formed of six Se atoms [Ugeda, M. et al. *Nature Phys* 12, 92–97 (2016)]. In Fig. R4(b), an adatom is observed as isolated, sharply bright atoms with a diameter of ~1 nm [Liebhaber, E. et al. *Nat Commun* 13, 2160 (2022)]. In Fig. R4(d), an inhomogeneous background gives rise to a modulation on the length scale of nanometers without long-range order [Lasek, K. et al. *ACS Nano* 14, 8473–8484 (2020)]. This inhomogeneous background comes from the disordered, subsurface intercalated atoms in bilayer intercalated Cr_xTe_y , and it is absent in the monolayer [Fig. R4(c)]. In our Fig. S2, the nanometer-scale disordered modulation falls into the category of an inhomogeneous background, rather than either adatoms or vacancies. The message we are trying to convey is that this inhomogeneous background can come from varying intensities at Nb intercalant sites beneath the surface.

Figure R4. Vacancies, adatoms, and inhomogeneous backgrounds in STM images. (a) The STM topographic image of monolayer NbSe_2 at $T = 25$ K, showing two isolated Se vacancies; setpoint -4 mV, 50 pA. Image reproduced from [Ugeda, M. et al. *Nature Phys* 12, 92–97 (2016)]. (b) The STM topographic image of a single Fe atom on NbSe_2 ; setpoint 4 mV, 200 pA. Image reproduced from [Liebhaber, E. et al. *Nat Commun* 13, 2160 (2022)]. (c, d) STM topographic images of monolayer and bilayer Cr_xTe_y films, respectively. Additional height modulations can be observed (d). Setpoint: (c) -10 mV, 0.8 nA; (d) -5 mV, 1.1 nA. Images reproduced from [Lasek, K. et al. *ACS Nano* 14, 8473–8484 (2020)]. (e) The STM topographic image of a $\text{Nb}_{1+x}\text{Se}_2$ film (sample NbSe180). Setpoint: -125 mV, 50 pA. (f) Gaussian convolution of (e) with a width of 15 pixels, revealing the background inhomogeneity. (e) and (f) are reproduced from our Figure S2.

Actions taken:

We have included this explanation in the text accompanying Fig. S2 (Page 4 in the Supporting information), detailing how vacancies, adatoms, and inhomogeneous distribution of subsurface intercalants should appear in STM topographic images.

“...STM images such as these ones can reveal the presence of various defects, including vacancies, adatoms, and intercalants. Vacancies would appear as missing bright spots in the hexagonal lattice, whereas adatoms would appear as distinct, large bright spots on top of the hexagonal lattice. What we observe are not vacancies or adatoms, but an inhomogeneous background to the hexagonal lattice, with brighter and darker patches.

....

For example, the area enclosed by the orange dashed line has a brighter intensity and may correspond to a region with more Nb intercalants, whereas the area enclosed by the green dashed line has a darker intensity and may correspond to a region with fewer Nb intercalants.”

Comment 5:

The STEM shows clear interlayer expansion when the vdW gap is filled with intercalations. I found this hard to understand. Intercalants will create covalent bonds [Nature 581, 171-177 (2020)], therefore the large weakly bonded vdW gap should be eliminated or partially eliminated. For instance, as an example from a sister family system, the c axis of VSe_2 (1:2, vdW gap fully empty) and VSe (1:1, vdW gap fully occupied) is 0.613 and 0.578 nm, respectively ($\sim -6\%$). How can we understand the lattice expansion in the case of Nb-Se?

Our response:

There are several studies of bulk $\text{Nb}_{1+x}\text{Se}_2$ that show that the interlayer distance expands with self-intercalation [e.g., Huisman, R. et al. *J. Less-Common Metals* 21, 187-193 (1970)]. However, this is not a general trend, as the reviewer noted, since the interlayer distance appears to shrink with self-intercalation in $\text{V}_{1+x}\text{Se}_2$. We presently speculate the following: whether the interlayer distance of $\text{M}_{1+x}\text{Se}_2$ expands or contracts upon self-intercalation depends on the ideal length of the quasi-covalent $M-M$ bond relative to the original interlayer spacing. Since V has smaller, localized $3d$ orbitals, it could be that the ideal V-V bond length is smaller and leads to a contraction of the interlayer spacing. Since Nb has larger $4d$ orbitals, it could be that the ideal Nb-Nb bond length is larger and leads to an expansion of the interlayer spacing. A recent work on

TaS₂ with larger *5d* orbitals reported that additional Ta atoms primarily fill the vdW gap resulting in substantial vdW gap expansion [Wu, S. et al. *Nano Lett.* 24, 1, 378–385 (2024)].

Actions taken:

We have modified the following sentence under the subsection “Stacking-selective self-intercalation”:

“XRD measurements over a macroscopic area consisting of many intercalated regions confirm an overall increase in the average interlayer spacing, consistent with previous polycrystal studies of Nb_{1+x}Se₂”

We clarify that our observed expansion of the interlayer distance is not a general trend, but simply consistent with what others have observed in literature for the Nb_{1+x}Se₂ system.

Comment 6:

The staggering of intercalants is quite interesting. Here it is only shown for the case of 7–8-layer films. Is the same for the rest of samples? What about the thicker films? Seems to me this staggering mechanism may be very well connected with the PLD growth process, as discussed in comment 2. The formation of intercalated-superstructures and twisting has been also reported for CVD-grown Cr-Te nanoplates [Nano Lett. 2021, 21, 9517–9525].

Our response:

In the thin films, the staggering phenomenon appears quite frequently in regions with 180° stacking of layers. However, as demonstrated by our new measurements on bulk Nb_{1+x}Se₂ single crystals, the observation of staggering is not universal.

Actions taken:

As detailed in our response to Comment 2, our new measurements on bulk single crystals and new DFT calculations have led us to revise our interpretation of the possible microscopic mechanism of stacking-selective self-intercalation.

Reviewer #2:

The authors present a thorough investigation of Nb_{1+x}Se₂ synthesized by PLD. They go beyond prior observations and demonstrations by correlating the self-intercalation with layer stacking, unambiguously identifying the nature and position of the intercalant, observing ARPES spectra that appears to be a mixture of 2Ha and 2Hb, reported on the electronic properties. This work is well written and free of errors and unjustified conclusions. I recommend publication as is.

Our response:

We thank the reviewer for their interest and for recommending our manuscript for publication. We are also grateful for the comments and suggestions.

Minor comment:

The authors state the photon energy of their monochromated Al K α source as 1486.6eV. While this is commonly reported in the literature, it is incorrect and likely a legacy of un-monochromated dual anode sources where the weighted average of the k-alpha1 (1486.7eV) and the k-alpha2 (1486.3eV) was used. Since a monochromator will be optimized to the maximum intensity the result will be a k-alpha1 line at 1486.7eV. While the incorrect 1486.6 eV is quite prevalent throughout the literature (and even some vendor websites), I don't think this error should be propagated to a nature journal. Unless the authors optimized their monochromator away from the maximum intensity then the photon energy would be 1486.7 eV.

The authors can see the work by J. Scheppe R.D. Deslattes T. Mooney and C.J. Powell as well as the LBL tables which both confirm the energy of the K α 1 line.

[https://doi.org/10.1016/0368-2048\(93\)02059-U](https://doi.org/10.1016/0368-2048(93)02059-U)

https://xdb.lbl.gov/Section1/Table_1-2.pdf

Our response:

We thank the reviewer for pointing this out. We fully agree with the reviewer that the photon energy might not always be correctly stated. Since we cannot measure or access the exact photon energy, that comes out of our X-ray monochromator, we have removed the absolute photon energy in the manuscript to avoid any confusions, or propagating the error. Instead, we only mention that we used monochromatized Al K α x-rays. The binding energy calibration is detailed in the Methods section.

Actions taken:

We have modified the descriptions in the Methods subsection “LEED, XPS, ARPES”

from

“The first system includes a SPECS ErLEED 150 instrument and a Kratos AXIS Ultra spectrometer with a monochromatized Al K_{α} source (1486.6 eV photon energy), with which we obtained LEED [Figs. S4(a) and S4(b)] and XPS data (Figs. S12 and S13).”

to

“The first system includes a SPECS ErLEED 150 instrument and a Kratos AXIS Ultra spectrometer with a monochromatized Al K_{α} source, with which we obtained LEED [Figs. S4(a) and S4(b)] and XPS data (Figs. S9 and S10).”

Optional suggestion:

When reviewing the SI, I noted that the results and interpretation of the XPS and XRD data is similar and consistent with those reported recently by Litwin et al (for $Nb_{1+x}Se_2$)[1] and Bonilla et al (for VSe_2)[2] and as such may warrant comparison. Since Bonilla et al also reported on the surface structure observed with STM it would be interesting to consider whether all observations are consistent.

<https://doi.org/10.1116/6.0002593>

<https://doi.org/10.1002/admi.202000497>

Our response:

We thank the reviewer for bringing these references to our attention. Litwin *et al.* observed via XRD a similar expansion of the interlayer distance in $Nb_{1+x}Se_2$ thin films grown by MBE, as well as the appearance of additional components in the Se $3d$ core level spectra in XPS. Bonilla *et al.* analyze the Se $3d$ core level spectra of $V_{1+x}Se_2$ and $Ti_{1+x}Se_2$ in the same way we do, assigning the component at higher binding energy to fourfold-coordinated Se. A point of distinction with Bonilla *et al.* is that in $V_{1+x}Se_2$ and $Ti_{1+x}Se_2$, their STM topographic images suggest that the intercalants form in-plane ordering, whereas in our $Nb_{1+x}Se_2$ thin films, our STM topographic images suggest that Nb intercalants are disordered.

Actions taken:

We have included the two references in the Supplementary Information, in the sections related to the XRD and XPS analysis.

Reviewer #3:

H Wang et al study the intercalation of excess Nb within hybrid PLD grown $Nb_{1+x}Se_2$ thin films, and they observe a strong interaction between the layer stacking order and intercalation. Using atomically resolved STEM imaging and EELS analysis, the authors find that their films contain both 180° and 0° stacking, and that intercalants preferentially occupy the octahedral sites between 180° stacked layers. Negligible intercalation was found between 0° stacked layers. The experimental data suggest that Nb intercalants energetically prefer the octahedral 180° sites, and this is confirmed via DFT calculations. The authors also argue that the presence of 0° stacking is driven by the inter-layer repulsion of Nb intercalants and a staging mechanism. The effects of intercalation on the electronic properties are investigated with ARPES and resistivity measurements. Both the intercalation of TMDs and the layer stacking in TMDs are topics of interest, and this manuscript presents an interesting system where intercalation and layer stacking cooperate in a non-trivial manner. The manuscript is well-written, and the characterization and analysis are complete. Below are some questions / comments for the authors.

Our response:

We thank the reviewers for their interest and positive assessment of our work, as well as their careful feedback.

Comment 1:

The term staging is most often associated with Li intercalation into graphite, which progresses from stage 4, to stage 3, stage 2, and finally stage 1, which corresponds to fully intercalated graphite. Do the authors propose that similar stages occur for $Nb_{1+x}Se_2$? Have different stages been observed experimentally? The authors should clarify their use of the term staging.

Our response:

We thank the reviewers for this comment. We indeed do not have snapshots of the thin films at different times during the growth, to really know whether the intercalation occurs in stages, as in Li intercalation of graphite. Therefore, we have revised our terminology from “staging” to “staggering,” simply to emphasize the phenomenon, which is a layer-by-layer modulation of the intercalant concentration we see in 180°-stacked regions, without making any implications of the microscopic process.

Actions taken:

We have replaced “staging” with “staggering” throughout the main text.

Comment 2:

The authors propose that the 0° layer stacking is driven by the inter-layer repulsion of Nb intercalation. So the 0° stacking forms to separate layers of fully intercalated 180° stacking. If this argument is accurate, then 0° stacking should only form in close proximity to intercalated 180° stacking. But the data in Fig. 3 seems to contradict this argument. In Fig 3d, the NbSe₂ film is entirely 0° stacked, except for a single layer of intercalated 180° stacking at the substrate interface. The topmost 0° stacked layers cannot be driven by an inter-layer Nb repulsion mechanism, since there is no Nb intercalation nearby. A similar situation is found in Fig. 3c.

Our response:

We thank the reviewers for pointing this out. After performing new control measurements on a series of Nb_{1+x}Se₂ single crystals grown via CVT, as well as new DFT calculations with larger supercells and lower *x*, we have revised our microscopic picture for stacking-selective self-intercalation. Please refer to our response to Comment 2 of Reviewer 1, or the revised Discussion section in the main text.

We no longer suggest that the 0° layers arise to separate heavily intercalated 180°-stacked layers. Instead, the 0°-stacked layers are favored when *x* is larger than 0.03–0.07, and the 180°-stacked layers are favored when *x* is greater than 0.25. What we missed earlier is that there are actually two polytype transformations in the Nb_{1+x}Se₂ phase diagram, from 180° to 0° stacking at around *x* = 0.03–0.07, and from 0° back to 180° stacking when *x* is greater than 0.25.

Actions taken:

Based on new experiments and calculations, we have updated our microscopic picture and interpretation of our previous thin-film results. New Figs. 5 and 6 have been added, and the Discussion section has been revised accordingly. We no longer propose that the 0° layers arise due to interlayer repulsion of Nb intercalants.

Comment 3:

Have the authors grown hPLD thin films of stoichiometric NbSe₂ (with no self-intercalation) and studied the layer stacking? Is it possible that the hPLD growth conditions yield mixed layer stacking even in the absence of self-intercalation?

Our response:

This is a good point. Unfortunately, we have not succeeded in synthesizing stoichiometric thin films of NbSe₂ thicker than one monolayer using our hPLD technique. Litwin *et al.* [Litwin, P. M. et al. *J. Vac. Sci. Technol. A* 41, 042707 (2023)] report the same difficulty with MBE – even when their Nb:Se beam-equivalent pressure ratio is 45000:1, their films are intercalated with *x* ~ 0.23.

Comment 4:

What is the lateral length-scale of the different layer stacking orders and intercalated layers? Do the authors have any wide field-of-view images showing changes in the layer stacking and intercalant structure? For instance, a lateral transition from 0° stacking to 180° stacking? Alternatively, do the authors have any selected area electron diffraction of the FIB'd samples, which would provide information on the layer stacking averaged across a large region?

Our response:

Figure R5 shows an atomic-resolution HAADF-STEM image of a region of the Nb_{1+x}Se₂ film with a lateral field of view of ~60 nm. The yellow box marks the position where a lateral transition from 0° stacking to 180° stacking occurs. The close-up image of the red box region evidences a simultaneous change in layer stacking and intercalation, showcasing the 180° stacking with filled vdW gaps (the upper-left part) and 0° stacking with relatively empty vdW gaps (the lower-right part).

From STM images (Fig. S2), we observe inhomogeneity in the lateral distribution of subsurface Nb intercalants on the order of several nanometers. From AFM images (Fig. S16), our film shows island and domain structures on the order of 100 nm.

We might estimate that the characteristic lateral length scale of different stacking and intercalation layers is on the order of 10–100 nm.

Figure R5. Atomic-resolution HAADF-STEM image of a region of $\text{Nb}_{1+x}\text{Se}_2$ film with a lateral length scale of around 60 nm. The yellow box marks the position where a lateral transition from 0° stacking to 180° stacking occurs. The close-up image of the red box region is shown in the bottom panel. The red dotted line marks the boundary between the 180° and 0° stacking regions.

Actions taken:

Figure R5 is now included as Fig. S11 (Page 12) in the Supplementary Information.

Comment 5:

For the ARPES measurements, what is the penetration depth? Is the entire film thickness being probed?

Our response:

The penetration depth of UV light is rather large compared to the inelastic mean free path of the photoelectrons. Therefore, what makes ARPES and XPS highly surface sensitive is not the penetration depth of the incoming light but the escape depth of the outgoing electrons. For electrons with small kinetic energies in the range of 15 eV (as is the case for ARPES) the inelastic mean free path (IMFP) is in the range of 1 nm [Seah, M. P. et al. *Surface and interface analysis*, 1(1), 2-11 (1979)]. Assuming that the electrons come from roughly 3 times the IMFP and a single layer of NbSe_2 is roughly 0.65 nm, most of the signal comes from the top 5 layers, which covers the majority of the entire film thickness.

Actions taken:

We have supplemented the descriptions of the probing depth of ARPES measurements in the Methods section (Page 15):

“Assuming that the photoelectrons in ARPES come from roughly three times the inelastic mean free path, which is approximately 1 nm for electrons with kinetic energies in the range of 15 eV⁵⁵, and an interlayer distance of roughly 0.65 nm, most of the signal comes from the top five layers, which covers the majority of the entire film thickness.”

Comment 6:

The text references a yellow arrow in Fig. 2f, but the arrow is pink. Also, the text refers to an orange arrow in Fig. 2i, but the arrow looks green.

Our response:

We thank the reviewers for pointing out the mismatch in the color descriptions of Fig. 2. We have conducted corresponding changes to make the text consistent with the color scales in Fig. 2f and 2i.

Actions taken:

We have modified the caption of Figure 2 (Page 6):

“...(h) is the background-subtracted EELS line scan at the Nb- $\text{N}_{2,3}$ edge taken along the vertical pink arrow in (f), perpendicular to the NbSe_2 layers...”

and the description of Figure 2 in the main text (Page 5):

“...For the intercalants, the onset, volume, and peak of the Nb- $\text{N}_{2,3}$ edge are shifted to lower energies [red, green, and black

arrows in Fig. 2(i),...”

Comment 7:

For the Nb oxidation state analysis, did the authors look at the fine structure of the Nb L-edge (Fig. 2b) to confirm their findings at the Nb N-edge? Does the N-edge offer advantages for chemical analysis compared to the L-edge?

Our response:

We thank the reviewers for the comments. Energy-loss fine structure analysis of L edges has been documented as a feasible way to determine the oxidation state of 3d transition metals, whose L edges occur at an energy loss below 1000 eV. However, previous studies show that a clear trend could not be observed for the fine structure of Nb-L edges (L_3/L_2 ratio, threshold energy) as a function of the Nb-oxidation state [Bach, D. et al. *Microsc. Microanal.* 15, 505–523 (2009)]. Additionally, the Nb- $L_{2,3}$ edges with a white-line shape that attributed to the transition of Nb $2p_{3/2}$ and $2p_{1/2}$ core electrons to unoccupied 4d and 5s states occur at high energy losses above 2370 eV and 2465 eV. EELS spectra of Nb- $L_{2,3}$ edges (Fig. 2b) have a rather poor signal-to-noise ratio compared to that of Nb-N edges occurs at low energy loss (34 eV) so that a slightly reduced positive valence state as suggested in this work is hard to be detected using Nb-L edges. Therefore, we solely employed the Nb-L edges to identify the atomic-scale distribution of Nb atoms in this work.

Comment 8:

In Fig. 1g, how were the error bars calculated?

Our response:

We thank the reviewer for this detailed comment. The Se-Nb-Se angle values in Fig. 1g are the mean values over each NbSe₂ layer. The error bar corresponds to the standard deviation of the Se-Nb-Se angle in each NbSe₂ layer. We added corresponding descriptions to the caption of Fig. 1 for clarification.

Actions taken:

We have added the description of the error bar to the caption of Fig. 1 (Page 4):

“...(g) A close-up HAADF-STEM image revealing the Nb_{1+x}Se₂ lattice structure (left) and the calculated interlayer spacing (middle) and Se-Nb-Se angle (right) averaged over each NbSe₂ layer. The error bars correspond to the standard deviations of the Se-Nb-Se angle in each NbSe₂ layer.”

REVIEWERS' COMMENTS

Reviewer #1 (Remarks to the Author):

In the revised version of the manuscript, the authors have addressed all my questions and concerns nicely. I would like to acknowledge the great efforts from the authors to conduct more experiments, calculations and revise the story and conclusions of the research. I think the present manuscript provides a more comprehensive understanding of the self-intercalation mechanism in the Nb-Se system and surely will be a great contribution for the community. Therefore, I can recommend its publication in its current form.

Reviewer #3 (Remarks to the Author):

The authors have addressed our questions and comments satisfactorily. Particularly, I am glad to see that the authors have carried out a substantial amount of extra work on CVT-grown bulk samples as well as additional DFT calculations. The revised manuscript is much improved.

Reviewer #4 (Remarks to the Author):

The authors addressed all of my prior comments, and the manuscript is greatly improved after the revisions. I believe the manuscript is suitable for publication in Nature Communications.

Based on the updated manuscript, I have a follow up question which the authors may consider.

The updated paper claims two stacking transitions as a function of x . For $x = 0$, the stacking is 180 degrees. For finite but small x , the stacking is 0 degrees. For large x (greater than 25 - 50%), the stacking switches back to 180 degrees. The hPLD grown sample with $x = 29\%$ is presumably near the second transition point. At this transition point, the energy of intercalated 180 degree stacked NbSe₂ is equal to the energy of intercalated 0 degree stacked NbSe₂ (From Figure 6a). Thus, one would expect phase coexistence between 180 degree stacking and 0 degree stacking, with each phase having similar amounts of intercalation. Experimentally, the authors observed phase coexistence, as expected. But they observe that nearly all of the intercalation is within 180 degree stacked regions, with no intercalation within the 0 degree stacked regions. I don't think this observation is adequately discussed or explained.

Based on their DFT calculations, have the authors considered the following: for a system with a fixed amount of intercalation (let's say 25% globally) and both 180 degree and 0 degree stacking (equal parts of each), where does the Nb intercalation go? Does it evenly disperse between the 180 and 0 degree

stacked regions, or does it preferentially occupy the 180 degree stacked regions? The extreme limit of this question could be evaluated by the following expression,

$$(E_{180\text{-degree-stacking_25\%-intercalation}} + E_{0\text{-degree-stacking_25\%-intercalation}}) - (E_{180\text{-degree-stacking_50\%-intercalation}} + E_{0\text{-degree-stacking_0\%-intercalation}})$$

If this expression is positive, it suggests that Nb intercalants would preferentially occupy the 180 degree stacked regions, leaving the 0 degree stacked regions intercalant-free.

Response to the Reviewers and Revisions

Reviewer #4:

The authors addressed all of my prior comments, and the manuscript is greatly improved after the revisions. I believe the manuscript is suitable for publication in Nature Communications.

Based on the updated manuscript, I have a follow up question which the authors may consider.

The updated paper claims two stacking transitions as a function of x . For $x = 0$, the stacking is 180 degrees. For finite but small x , the stacking is 0 degrees. For large x (greater than 25 - 50%), the stacking switches back to 180 degrees. The hPLD grown sample with $x = 29\%$ is presumably near the second transition point. At this transition point, the energy of intercalated 180 degree stacked NbSe₂ is equal to the energy of intercalated 0 degree stacked NbSe₂ (From Figure 6a). Thus, one would expect phase coexistence between 180 degree stacking and 0 degree stacking, with each phase having similar amounts of intercalation. Experimentally, the authors observed phase coexistence, as expected. But they observe that nearly all of the intercalation is within 180 degree stacked regions, with no intercalation within the 0 degree stacked regions. I don't think this observation is adequately discussed or explained.

Based on their DFT calculations, have the authors considered the following: for a system with a fixed amount of intercalation (let's say 25% globally) and both 180 degree and 0 degree stacking (equal parts of each), where does the Nb intercalation go? Does it evenly disperse between the 180 and 0 degree stacked regions, or does it preferentially occupy the 180 degree stacked regions? The extreme limit of this question could be evaluated by the following expression,

$$(E_{180\text{-degree-stacking}_{25\text{-intercalation}} + E_{0\text{-degree-stacking}_{25\text{-intercalation}}}) - (E_{180\text{-degree-stacking}_{50\text{-intercalation}} + E_{0\text{-degree-stacking}_{0\text{-intercalation}}})$$

If this expression is positive, it suggests that Nb intercalants would preferentially occupy the 180 degree stacked regions, leaving the 0 degree stacked regions intercalant-free.

Our response:

We thank the reviewer for raising this important comment. What we observed is not merely a nanoscale phase coexistence of 180°- and 0°-stacked regions with comparable amounts of intercalation, but a phase separation into 180°-stacked regions with high concentration of intercalants and 0°-stacked regions with low concentration of intercalants. We suspect that spinodal decomposition is at play: namely, near a global $x \sim 0.29$, the second partial derivative of the free energy with respect to the intercalant concentration is negative. A system with uniform $x = 0.29$ can then spontaneously gain energy via phase segregation into $x > 0.29$ regions with 180° stacking and $x < 0.29$ regions with 0° stacking.

We have not performed enough calculations at this point to say whether DFT supports a picture of spinodal decomposition. We find that the combined energy per formula unit of a 180°-stacked, 50%-intercalated compound and 0°-stacked, 0%-intercalated compound is actually higher than the combined energy per formula unit of a 180°-stacked, 25%-intercalated compound and a 0°-stacked, 25%-intercalated compound. This limit is likely too extreme, as we know there is another stacking transition around $x = 0.03\text{--}0.07$. We probably need to compare the energies of the $x = 0.25$ structures with those of the $x = 0.25 \pm \delta$ structures, where δ is much smaller than 0.25, or perform molecular dynamics simulations.

Actions taken:

We have modified the following parts in the Discussion. Page 10:

“Our STEM images disclose that epitaxial films of Nb_{1+x}Se₂ near a global $x \sim 0.29$ exhibit a nanoscale **separation into** 180°-stacked layers with a large number of Nb intercalants, perhaps several tens of percentage occupancy (considering the average x value), and 0°-stacked layers whose number of Nb intercalants falls close to the detection limit.”

Pages 13:

“Within the parameter space of our growth conditions, we only reached the boundary of the second transition around $x \sim 0.29$ in the thin films, where we observed both 0° and 180° -stacked layers coexisting on the nanoscale. Here, the intercalants are not homogeneously distributed, but preferentially reside in the 180° -stacked layers, while leaving the 0° -stacked layers much emptier in comparison. The stacking selectivity of the intercalants points to a phase separation mechanism, perhaps spinodal decomposition, where a homogeneously intercalated $x \sim 0.29$ phase is unstable against nanoscale segregation into 180° -stacked layers with local $x > 0.29$ and 0° -stacked layers with local $x < 0.29$ [Fig. 6(c)].”

Additional note:

We fixed some minor typos:

Pages 11:

“At 0% intercalation, the total energy after structural relaxation of 180° -stacked bilayer NbSe₂ is only **6 meV** per formula unit smaller than that of 0° -stacked bilayer NbSe₂ [Fig. 6(a)]”

Previously, we reported 2 meV, but this is the energy per atom. Figure 6(a) has also been updated to reflect 6 meV per formula unit.

SI, Fig. S14 caption:

“(a) The difference in DFT-computed total energy per **atom**, ΔE_{tot} , of pristine 0° -stacked bilayer NbSe₂ relative to 180° -stacked bilayer NbSe₂.”

The values here are the total energy per atom, not per formula unit, as previously written.